# Disrupting upstream translation in mRNAs is associated with human disease

David S. M. Lee [1,2], Joseph Park [1,2,3], Andrew Kromer[1], Aris Baras [4], Daniel J. Rader [1,3,5], Marylyn D. Ritchie [1,2], Louis R. Ghanem[6,7,9✉] & Yoseph Barash [1,5,8✉]

Ribosome-profiling has uncovered pervasive translation in non-canonical open reading frames, however the biological significance of this phenomenon remains unclear. Using genetic variation from 71,702 human genomes, we assess patterns of selection in translated upstream open reading frames (uORFs) in 5′UTRs. We show that uORF variants introducing new stop codons, or strengthening existing stop codons, are under strong negative selection comparable to protein-coding missense variants. Using these variants, we map and validate gene-disease associations in two independent biobanks containing exome sequencing from 10,900 and 32,268 individuals, respectively, and elucidate their impact on protein expression in human cells. Our results suggest translation disrupting mechanisms relating uORF variation to reduced protein expression, and demonstrate that translation at uORFs is genetically constrained in 50% of human genes.

[1] Department of Genetics, Perelman School of Medicine, University of Pennsylvania, Philadelphia, PA, USA. [2] Institute for Biomedical Informatics, Perelman School of Medicine, University of Pennsylvania, Philadelphia, PA, USA. [3] Department of Medicine, Perelman School of Medicine, University of Pennsylvania, Philadelphia, PA, USA. [4] Regeneron Genetics Center, Regeneron Pharmaceuticals, Tarrytown, NY, USA. [5] Institute for Translational Medicine and Therapeutics, Perelman School of Medicine, University of Pennsylvania, Philadelphia, PA, USA. [6] Division of Gastroenterology, Hepatology and Nutrition, The Children's Hospital of Philadelphia, Philadelphia, PA, USA. [7] Department of Pediatrics, Perelman School of Medicine, University of Pennsylvania, Philadelphia, PA, USA. [8] Department of Computer and Information Science, School of Engineering, University of Pennsylvania, Philadelphia, PA, USA. [9]Present address: Janssen Research and Development, Spring House, PA, USA. ✉email: lghanem@its.jnj.com; yosephb@upenn.edu

The classic view of information processing in the cell by gene expression occurs through transcription followed by translation. This basic flow is often complicated by regulatory elements which confer additional stages of processing and control. In particular, upstream open reading frames (uORFs) are segments of 5′UTR mRNA sequences that can initiate and terminate translation upstream of protein-coding start codons. Specific uORFs are known to control protein expression by tuning translation rates of downstream protein-coding sequences, and potential uORFs have been identified in ~50% of all human protein-coding genes[1,2].

Translation initiation is the rate-limiting step controlling post-transcriptional gene expression[3], and rates of translation initiation can significantly impact mRNA stability[4–8]. Cap-dependent translation initiation begins when the 40 s ribosomal subunit encounters a start codon as it scans along the 5′UTR. At the start codon, peptide synthesis initiates when the 40 s subunit acquires the 60 s subunit with other translation initiation factors. If scanning ribosomes encountering uORFs prematurely initiate translation in the 5′UTR, upon reaching the uORF termination codon the ribosome may dissociate from the mRNA transcript, or the 40 s subunit may resume scanning after the 60 s subunit is lost. Resumption of scanning leads to translation of downstream reading frames only if the necessary translation initiation factors are reacquired by the 40 s subunit before reaching the downstream start codon. Thus, the spatial combination of uORFs and protein-coding start codons can produce different effects on downstream protein translation.

Previous analyses of large-scale population data have shown that genetic variants creating new uORFs are rare, suggesting that these variants are subjected to strong negative selection due to their capacity to cause pathogenic loss-of-function of associated proteins[2,9]. Moreover, it has been shown that variants destroying stop codons in translated uORFs are under strong negative selection, presumably because the resultant translational read-through can decrease start codon recognition and translation initiation at the coding sequence (CDS)[10]. In contrast, less is known about the impact of genetic variation within translated uORFs. Furthermore, recent untargeted ribosome-profiling experiments have revealed striking evidence of active translation at thousands of uORFs throughout the genome, but the biological significance of this phenomenon remains unresolved[2].

Here we use translated uORFs mapped through ribosome-profiling experiments and a deep catalog of human genetic variation to characterize patterns of selection acting on single nucleotide variants (SNVs) in uORF sequences. We assess evidence for the functional importance of translation at uORFs, and explore possible phenotypic consequences associated with genetic variation in these sequences. Using the allele frequency spectrum of SNVs from 71,702 whole genome sequences in gnomAD, we find that SNVs introducing new stop codons, or creating stronger translation termination signals in uORFs are under strong selective constraints within 5′UTRs. We propose that these variants are under selective pressure because they disrupt translation initiation at downstream protein-coding sequences. We then utilize the Penn Medicine Biobank (PMBB) to discover new, robust disease-gene associations using uORF stop-creating and stop-strengthening variants and replicate these associations in the UK Biobank (UKB), and by gene burden tests aggregating rare protein-coding loss-of-function variants. Finally we validate the impact of uORF stop-creating and stop-strengthening variants on protein expression for our top phenome-wide significant associations. These data demonstrate that genetic variants in translated uORFs that create new stop codons, or strengthen existing stop codons can contribute to disease pathology by changing protein expression.

## Results

**Variants introducing new stop codons in uORFs are under strong negative selection.** Since elongating ribosomes must translate uORFs before they reinitiate translation at the CDS, we hypothesized that genetic variants introducing new stop codons in translated uORFs could impede downstream translation initiation. Because these variants interrupt translation without affecting the CDS directly, we term them upstream termination codons (UTCs) to distinguish them from premature termination codons within protein-coding sequences.

To estimate the deleteriousness of UTC variants, we assessed their frequency spectrum in gnomAD using the mutability-adjusted proportion of singletons (MAPS) metric. MAPS compares the strength of selection acting against different classes of functional variation by assessing the relative enrichment for rare singleton (one sequenced allele) variants in gnomAD, adjusted for local mutation rates (see methods). More deleterious groups of SNVs—including premature termination codons and essential splice site mutations—show greater enrichment in singletons in gnomAD, and consequently have higher MAPS scores. MAPS has previously been used to assess patterns of selective pressures acting on different classes of variation in both protein-coding and non-coding regions of the genome[9,11–15].

Using translated uORFs from 4392 genes identified by deep ribosome profiling of two human cell lines (Supplementary Fig. 1)[16], we mapped genetic variation from 71,702 whole-genome sequences in gnomAD (version 3)[12]. We identified the subset of UTC variants by selecting SNVs which mutated uORF codons to either UGA, UAG, or UAA in the mapped uORF reading frame (Fig. 1a). We calculated MAPS scores for these UTC variants, finding that they are under strong negative selection within 5′UTRs, comparable to that of missense mutations in canonical protein-coding regions of the genome (Fig. 1b). Indeed, MAPS scores for these variants are significantly higher than all uORF variants (Fig. 1b, $P < 0.001$), sets of uORF variants matched by their underlying trinucleotide mutation context (Supplementary Fig. 2, $P < 0.001$—see methods), all 5′UTR variants creating UTCs outside of mapped translated uORFs ($P = 0.0441$), and stop-creating variants in non-canonical ORFs (ncORFs) from 3′UTRs, translated pseudogenes, and lncRNAs mapped by ribosome-profiling from the same study (Fig. 1b $P = 0.0041$, Supplementary Fig. 1). Intriguingly, MAPS scores were highest for variants predicted to introduce strong (UAA) stop codons that are less susceptible to translational read-through[17–19]. In contrast, variants introducing the weaker UGA stop codon exhibited MAPS scores that are only nominally higher than all uORF variants ($P = 0.2833$), suggesting that they may be less deleterious by comparison. To account for the possibility that the heightened MAPS scores for UTC variants resulted from overlap between 5′UTRs and annotated coding sequences in different mRNA isoforms, we repeated this analysis excluding all uORF variants overlapping with any annotated CDS sequence. Re-calculated MAPS scores with all CDS-overlapping variants removed remained essentially unchanged (Supplementary Table 1), ruling out the possibility that the enrichment in rare variation for UTC variants is driven by selection on coding sequences. Additionally, we previously observed that variants destroying the central guanine of putative G-quadruplex forming sequences exhibit heightened MAPS scores in UTRs[14]. We repeated this analysis with all potential G-quadruplex disrupting variants ($n = 57$) excluded, seeing a negligible effect on MAPS scores for all UTC variants (MAPS = 0.0377, 95% CI: 0.0196–0.0557). Overall, the strong selective pressure to remove UTC variants implies that these variants are also more likely to have functional biological consequences.

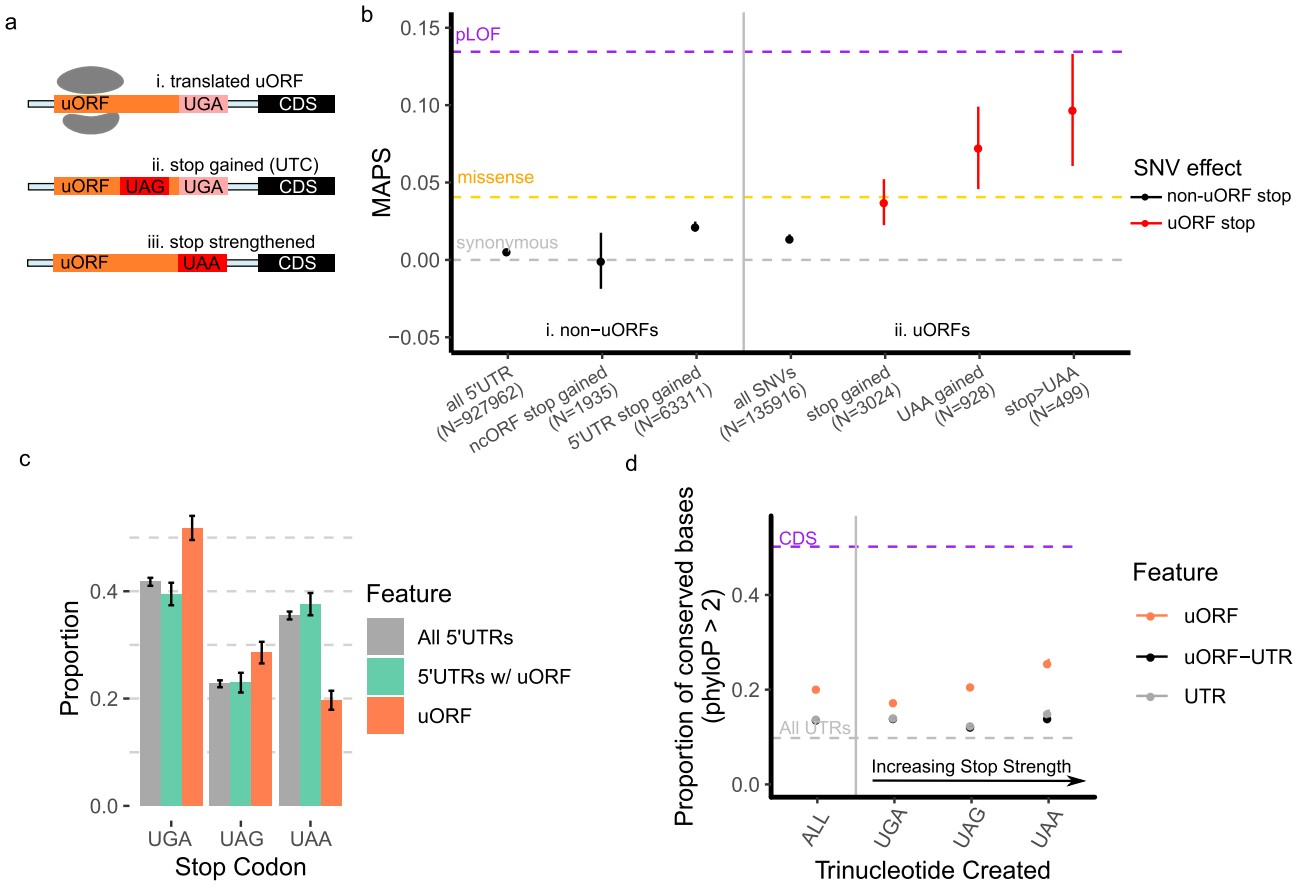

**Fig. 1 Stop-introducing and stop-strengthening variants in translated uORFs are under strong negative selection. a** Examples of possible stop-gained (UTC upstream termination codon) or stop-strengthened variants in translated uORFs. **b** Points showing mutability-adjusted proportion of singletons (MAPS) scores for different classes of stop-introducing variants in translated uORFs. Gray, orange, and purple dashed lines represent MAPS scores for synonymous, missense, and predicted loss-of-function (pLOF) SNVs affecting canonical protein-coding sequences in gnomAD. (i) MAPS scores for non-uORF variants including all 5′UTR SNVs, stop gained variants in ncORFs in 3′UTRs, lncRNAs, and pseudogenes), and all 5′UTR stop gained variants. (ii) MAPS scores for all uORF SNVs and stop gained variants in uORFs show that uORF UTC variants are significantly enriched for singletons. This is also observed for UAA-creating, and stop-strengthening SNVs in translated uORFs. Error bars represent bootstrapped 90% confidence intervals. **c** Relative frequencies of trinucleotides used as uORF stop codons compared to untranslated regions of uORF-containing 5′UTRs, or all 5′UTRs shows uORFs are significantly enriched for weaker (UGA, UAG) stop codons and depleted of the UAA stop codons compared to control sequences. Mean proportion over $n = 10,000$ random samples with replacement are represented by columns. Error bars represent 95% bootstrapped confidence intervals. **d** Points representing the proportion of strongly conserved (phyloP >2) bases by phyloP scores from 100-way vertebrate alignments for uORF stop-creating, non-uORF stop-creating in uORF-containing UTRs, and non-uORF stop-creating in all UTR genomic positions. Error bars represent 90% bootstrapped confidence intervals.

**Translated uORFs use weak stop codons**. Stop codons have different translation termination efficiencies in both prokaryotes and eukaryotes, with the hierarchy following the general pattern of UAA > UAG > UGA[17,20,21]. Given the observed selection against UTC variants in translated uORFs, and in particular against UAA-introducing variants, we next asked whether stop codon usage by translated uORFs is distinct from the background distribution of UGA, UAG, and UAA trinucleotides in 5′UTRs. To perform this comparison, we determined the relative frequency that UGA, UAG, or UAA trinucleotide sequences appeared within non-translated 5′UTR sequences, and compared this frequency to the distribution of stop codons used in translated uORFs. To further control for the possibility that translated-uORF containing UTRs might have significantly different background nucleotide distributions, we also assessed the relative frequency of UGA, UAG, or UAA trinucleotides from uORF-containing UTRs with translated uORF sequences excluded. Strikingly, we find that translated uORF stop codons are significantly depleted of UAAs compared

to background UTR distributions (Fig. 1c), suggesting that weaker stop-codons (UGA, UAG) are preferred (permutation $P < 0.001$ compared to all UTRs, $P < 0.001$ compared to uORF-containing UTRs). By comparison, stop codon usage in canonical protein-coding regions of the genome decreases from UGA to UAA to UAG[22], suggesting that the relative depletion of UAA stop codons in Fig. 1c is specific to uORFs. Indeed there are approximately 45% less uORF UAA stop codons compared to the relative frequency of UAA trinucleotides in adjacent untranslated UTR sequences (uORF-UAA = 19%, matched UTR-UAA = 35%—Supplementary Table 2). In contrast, UGA stop codons are enriched in translated uORFs compared to non-translated UTR sequences (permutation $P < 0.001$ compared to all UTRs, $P < 0.001$ compared to uORF-containing UTRs).

Given the depletion of UAA-stop codons in translated uORFs, we next asked whether variants changing weaker stop codons (UGA, UAG) to UAA were also enriched for singletons. Compared to synonymous and missense variation within the

protein-coding genome, we find that the MAPS metric for stop-strengthening variants is significantly higher (Fig. 1b-ii). This difference remained significant compared to uORF variants matched by trinucleotide context, indicating that this effect is specific to uORF stop codons ($P = 0.012$, Supplementary Fig. 2). Given that UAA codons can facilitate greater termination efficiency and more rapid ribosomal dissociation from mRNAs compared to UAG and UGA codons[17,23], these results are consistent with the possibility that stronger stop codons in uORFs can also increase the efficiency of translation termination in the 5′UTR. Thus, like UTC variants, stronger stop codons in uORFs may be disfavored because they decrease the probability that ribosomes reinitiate translation at downstream coding sequences.

**Genomic positions that can create new stop codons in uORFs are conserved.** Since the power of MAPS estimates are limited by the number of variants observed in gnomAD, we assessed the evolutionary conservation of each possible uORF stop-creating position as complementary evidence for their functional significance. For this, we compared the distribution of phyloP scores across potential uORF-stop-creating positions derived from the UCSC 100-way phyloP vertebrate alignment[24]. Specifically, for each potential new stop site, we compared the proportion of genomic positions with a phyloP score of >2—corresponding to strong conservation across multi-vertebrate alignment— versus those positions that were not strongly conserved (phyloP <2). A similar approach has been used to show that genomic positions with the potential to produce new uORFs are strongly conserved across vertebrates[9].

We performed several assessments of phyloP scores across 5′UTR contexts. Consistent with our MAPS analysis, potential stop-creating positions in translated uORFs are also more likely to be conserved compared to UTR positions matched by distance to the downstream CDS. This difference remained significant even when compared to potential stop-creating positions in 5′UTR sequences adjacent to (but not within) translated uORFs (Fig. 1d). Strikingly, conservation at each stop-creating position within mapped translated uORFs mirrored the strength of stop-codon contexts, with a positive correlation between the strength of the potential stop codon introduced and the proportion of uORF genomic positions that are conserved. This trend was not observed for non-translated 5′UTR contexts (Fig. 1d). In all cases, the proportion of conserved bases for each class of potential stop-creating variant was significantly higher than those positions in all 5′UTRs, and particularly within untranslated regions of translated-uORF containing UTRs ($P < 0.001$, Fig. 1d). Moreover, the proportion of highly conserved bases at possible stop-creating positions increased in association with increasing gene constraint, as determined by the gnomAD LOEUF score, and remained significantly higher than non-uORF 5′UTR stop-creating positions (Supplementary Fig. 3). Together, these complementary analyses support our initial findings that UTC variants are under strong negative selection within the human genome, and further strengthens the evidence that UTC variants may functionally disrupt protein expression.

**uORFs are not under strong selection to maintain amino acid identity.** Multiple transcriptome-wide ribosome profiling studies have proposed that some uORFs can encode functional micropeptides with important cellular roles[16,25,26]. This has fostered significant interest in the possibility that translated, ncORFs represent an overlooked class of potentially functional micropeptides with biological activity independent of the downstream protein-coding sequences[26,27]. If many uORFs encoded functional micropeptides, the pattern of constraint against UTC variants

might also reflect selection to preserve micropeptide function rather than downstream translation initiation. To address this possibility, we asked whether uORFs broadly exhibit similar constraints against missense variation, compared to known protein-coding regions of the genome, that could imply peptide functionality. We compared MAPS scores for predicted missense versus synonymous mutations in translated uORFs to those in canonical protein-coding regions of the genome (Fig. 2a). The MAPS scores for missense mutations in uORFs were significantly lower than that of missense mutations in canonical protein-coding regions of the genome, and not significantly higher than MAPS scores for synonymous variants in translated uORFs ($P = 0.7118$, Fig. 2a-iv). These results indicate that selection to maintain amino acid identity in uORF-encoded micropeptides is weak compared to canonical protein-coding sequences. As an additional control, we computed MAPS scores for predicted missense and synonymous mutations in 693, 1188, and 276 translated ncORFs mapped by ribosome profiling in 3′UTRs (dORFs), long-noncoding RNAs, and pseudogenes respectively, as these sequences are not thought to broadly encode for functional peptides. Similar to uORFs, predicted missense variants in these additional ncORFs were not significantly higher than predicted synonymous variants by MAPS score (dORFs $P = 0.3532$; lncRNAs $P = 0.7777$, pseudogenes $P = 0.4523$, Fig. 2a-i-iii).

Since many translated uORFs are short, we asked whether longer uORFs might exhibit greater selection against missense variants compared to shorter uORFs. To test this possibility, we divided uORFs into long sequences >118 codons comprising the top 25% longest mapped uORFs, and short uORFs <118 codons in length. MAPS scores for missense variants in long versus short uORFs yielded no evidence of significant constraint acting on amino acid changing variants compared to synonymous SNVs (long uORFs $P = 0.178$, short uORFs $P = 0.9628$, Fig. 2a-v).

Surprisingly, we observed that MAPS scores for both synonymous and missense variants in translated uORFs deviated significantly from all 5′UTR variation (Fig. 2a-iv). These heightened MAPS scores implied that uORF variants are under increased negative selection compared to all 5′UTR variants. The absence of similar effects for variants in dORFs, lncRNAs, or translated pseudogenes implies that this enrichment in singletons is unique to translated uORFs. One possibility is that synonymous variation in uORFs reflect selective pressures to maintain translational efficiency by preserving codon optimality. Messenger RNAs that are enriched with more optimal codons are both more stable, and more efficiently translated by ribosomes[28]. Like UTC variants, uORF mutations introducing suboptimal codons could therefore slow translational elongation and impede downstream translation initiation at the CDS. Indeed, variants introducing suboptimal codons in translated uORFs have been shown to disrupt translation initiation at downstream coding sequences[29–31], and more generally 5′UTRs are under selective pressures to maintain their capacity for facilitating translation initiation at the CDS[32,33].

To test whether translated uORFs are constrained to maintain codon optimality, we asked if MAPS scores for SNVs predicted to decrease codon optimality differed from those that increased codon optimality (Fig. 2b). Using experimentally-determined codon-stability coefficients (CSCs)[34], we matched each uORF SNV with its predicted consequence to codon optimality, and compared MAPS scores for optimality-increasing versus optimality-decreasing SNVs. As expected, SNVs increasing codon optimality were indistinguishable from all 5′UTR variants ($P = 0.1929$, Fig. 2c). In contrast, variants predicted to decrease codon optimality had significantly higher MAPS scores ($P < 0.001$), although the magnitude of this difference is moderate compared to UTC variants (Fig. 1b). This effect remained significant

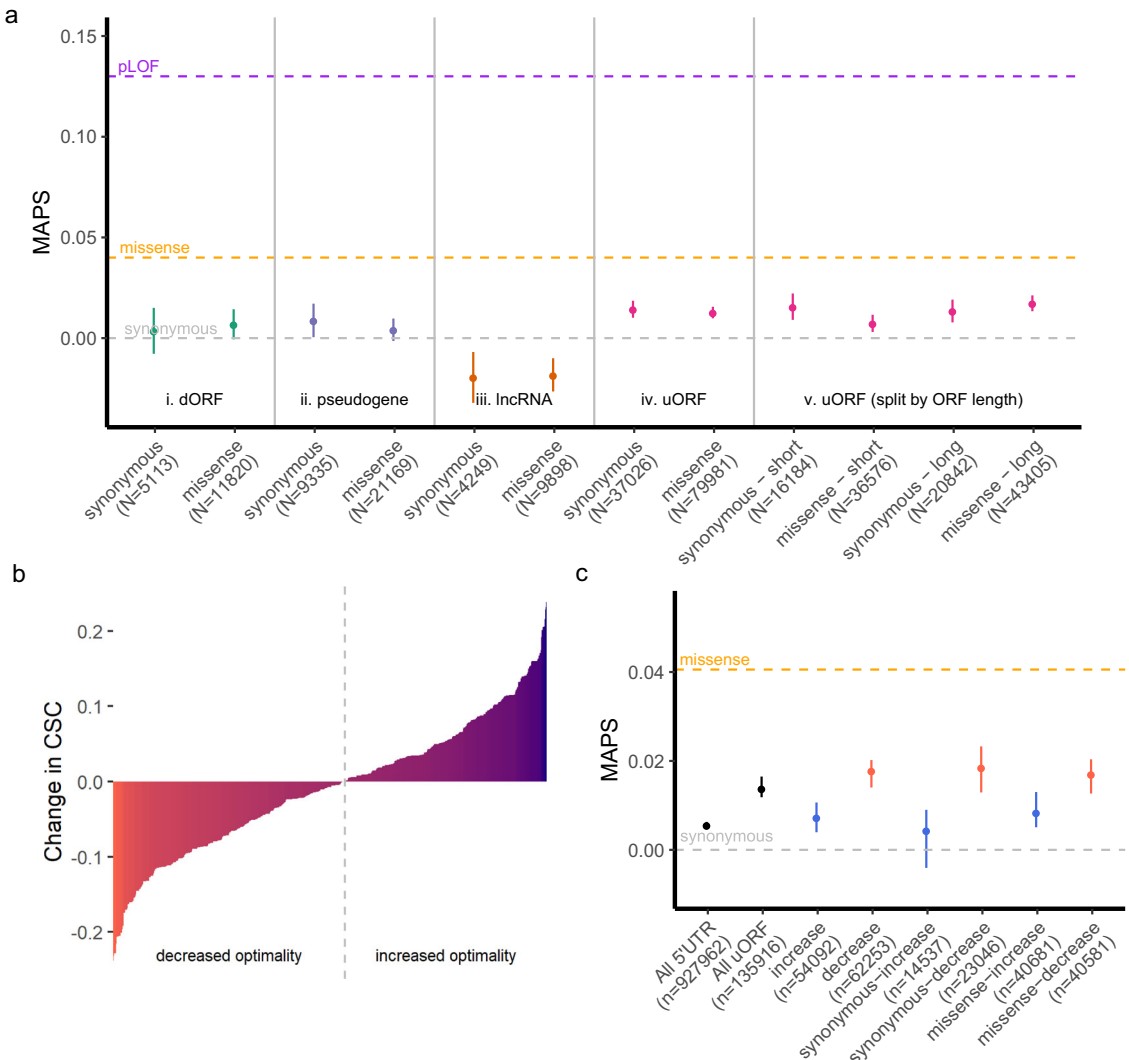

**Fig. 2 uORFs do not exhibit strong selective pressure to maintain amino acid identity. a** Points representing MAPS scores for single nucleotide variants within each ncORF category separated by predicted consequence (synonymous or missense) in each ORF. (i–iv) Allele frequencies for predicted missense SNVs are not significantly enriched for singletons than those for predicted synonymous SNVs. (v) MAPS scores are no different for long uORFs (>118 codons) compared to the rest (short). Gray, orange, and purple dashed lines represent MAPS scores for synonymous, missense, and predicted loss-of-function (pLOF) SNVs affecting canonical protein coding sequences in gnomAD. Error bars represent bootstrapped 90% confidence intervals. **b** Translated uORF variants ranked by predicted change to codon optimality using codon stability coefficient (CSC) scores from SLAM-seq (red = decreasing, blue = increasing)[34]. Gray dotted line denotes boundary separating optimality increasing versus decreasing SNVs. **c** Points representing MAPS scores for SNVs separated by predicted consequence on codon optimality shows heightened constraint against decreasing optimality variants, while variants increasing optimality are indistinguishable from all 5′UTR variants. Error bars represent bootstrapped 90% confidence intervals.

regardless of whether variants were predicted to cause synonymous or missense mutations ($P = 0.0125$ for synonymous; $P = 0.009$ for missense), and was notably absent for translated ORFs in 3′UTRs, lncRNAs, and pseudogenes (Fig. 2c and Supplementary Fig. 4). Furthermore, this pattern of increased constraint against optimality-decreasing variants was robust to the use of CSC scores derived from alternative experimental approaches across several cell lines (Supplementary Fig. 5)[34]. Together, these observations further support the hypothesis that natural selection acts to maintain the capacity for translational initiation at downstream coding sequences by preserving translational elongation efficiency in uORFs.

**uORF start codons are conserved and under strong selective pressure.** The finding of heightened selection against translation-

interrupting variants in uORFs raises the question of why translated uORFs continue to persist in a large fraction of human genes. Evidence that uORF-CDS organization, and the strength of uORF repression is strongly conserved across vertebrates, suggests that translation at uORFs is maintained to regulate downstream translation initiation[35]. Moreover, variants destroying uORF start codons have been implicated in the development of cancer[36]. To provide further genetic evidence that translation at uORFs is maintained by selection, we asked whether allele frequencies for variants affecting uORF start codons also exhibited strong selection to maintain their capacity for translation initiation. Using the MAPS metric, and genome-wide phyloP scores, we evaluated patterns of variation affecting uORF start codons. Since many translated uORFs begin with non-canonical start codons (Fig. 3a-i), we distinguish between variants maintaining the start context by affecting the first position of the NUG trinucleotide from those that disrupt translation

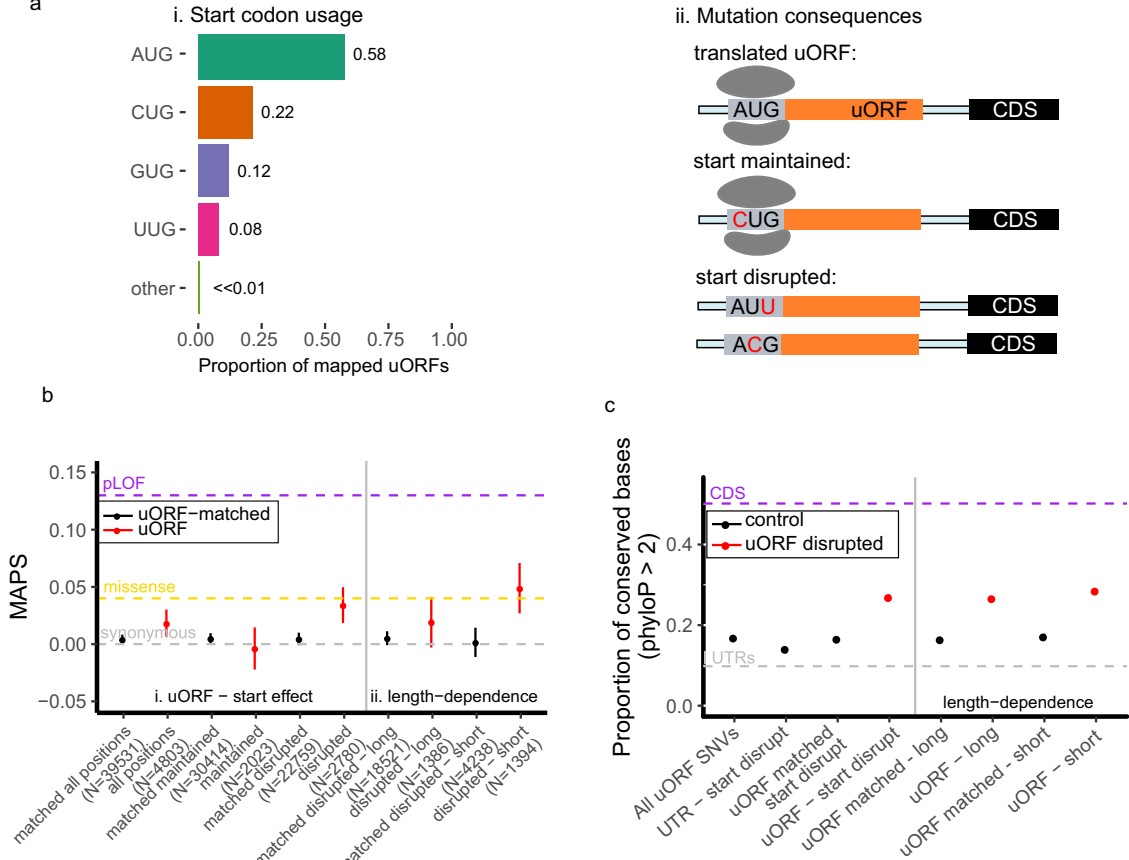

**Fig. 3 Selective pressure to preserve uORF start codons. a** (i) Distribution of start codon usage for experimentally mapped translated uORFs and (ii) possible consequences of variants affecting uORF start codons. **b** Points representing (i) MAPS scores for start-disrupting SNVs are compared to uORF variants matched by trinucleotide mutation context. (ii) Start-disrupting SNVs for short (<20 codons) uORFs are under stronger negative selection compared to start-disrupting variants for long (≥20 codons) uORFs. Error bars represent bootstrapped 90% confidence intervals. **c** Points representing proportion of conserved bases for possible start codon disrupting positions in uORFs compared to all uORF SNVs, UTR-matched start-disrupting positions, and uORF-matched start-disrupting positions in translated uORFs. Start-disrupting genomic positions of short uORFs are more strongly conserved by phyloP scores compared to matched start-disrupting positions within uORFs. Error bars represent bootstrapped 90% confidence intervals.

initiation by mutating the last two nucleotides in the uORF start codon (Fig. 3a-ii). As expected, start-maintaining variants are no more enriched for singletons in gnomAD compared to synonymous protein coding variants. In contrast, start-disrupting variants are enriched for singletons at a level comparable to that of protein-coding missense and UTC variants (Fig. 3b). The heightened pressure to maintain translational initiation at uORF start codons is similarly reflected in phyloP scores for uORF start-disrupting genomic positions compared to distance-matched UTR controls ($P$ < 0.001), and uORF-matched controls ($P$ < 0.001, Fig. 3c). These data show that translation initiation at uORFs is evolutionarily constrained in humans, and are consistent with previous reports that uORF start codons are frequently conserved across species.

Taken together, our analyses of genetic variation in gnomAD show enrichment for rare allele frequencies in the frequency spectra of uORF start-disrupting, stop creating, and stop-strengthening variants. Results from our analyses indicate that these classes of variation are under a heightened degree of negative selection, and imply that processes of translation initiation, elongation, and termination at translated uORFs are maintained by selective pressure.

**uORF-disrupting variants associate genes with new disease phenotypes.** The heightened MAPS score for UTC variants

suggests that they are also likely to be functional. To explore the possibility that UTC and uORF stop-strengthening variants might contribute functionally to human disease susceptibility, we performed a phenome-wide association study (PheWAS) of predicted uORF-disrupting variants using the PMBB—a large academic biobank with exome sequencing linked to EHR data for 10,900 individuals[37].

Using exome sequencing from the PMBB, we identified heterozygous and homozygous individuals carrying UTC and stop-strengthening variants. For the former class, we focused on variants introducing UAA stop codons, as the heightened MAPS score for such variants implied these variants would be most deleterious. Filtering for variants with at least five heterozygous carriers with high-quality genotype, we identified ten variants matching the above criteria (six stop-strengthening variants, four UAA-UTC variants). For each of these variants we performed a single-variant PheWAS across 800 EHR phenotypes. Of those ten candidates, six passed an FDR threshold of 0.1 ($P$ < 1.25e−4) used in previous PheWAS studies[38,39], including 5/6 of the stop-strengthening variants and 1/4 of the UAA UTCs. Even more strikingly, two of these six variants passed a highly-conservative Bonferroni correction ($P$ < 6.25e−6), both being uORF stop-strengthening variants. The stop-strengthening variant in *PMVK* was associated with increased risk of Type 1 diabetes while the stop-strengthening variant in *VPS53* was associated with a

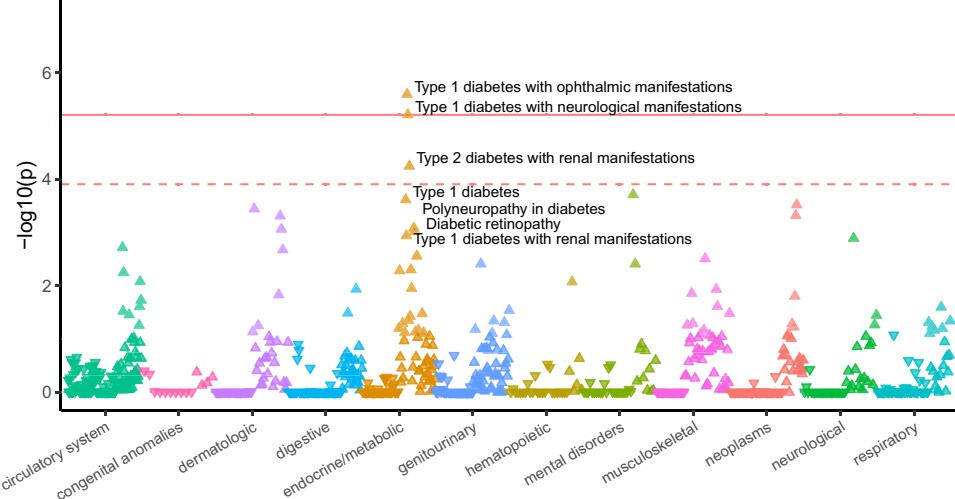

**Fig. 4 Phenome-wide association study (PheWAS) of predicted stop-strengthening variant in a translated uORF in PMVK.** PheWAS plot of translated uORF stop-strengthening variant in the 5′UTR of PMVK (N = 65 carriers) in the Penn Medicine BioBank. ICD-9 and ICD-10 Phecodes are organized and plotted by category on the X-axis. Each point represents a single Phecode. The height of each point corresponds to the −log$_{10}$(P value) of the association between the variant and Phecode using a logistic regression model adjusted for age, age$^2$, sex, and the first ten principal components (PCs) of genetic ancestry. The solid red line represents the threshold for Bonferroni-adjusted significance (P = 6.25e−6) and the red dashed line represents the FDR < 0.10 threshold (P = 1.25e-4) accounting for multiple comparisons. The direction of each arrowhead corresponds to increased risk (up) or decreased risk (down).

protective effect against anxiety disorders (Fig. 4, Supplementary Fig. 6, Table 1, and Supplementary Table 3). Notably, of the identified phenotype-associated variants, only VPS53 is annotated as a *cis*-eQTL in the latest GTEx release (version 8).

**Replication of novel associations in UKB.** To replicate associations from our exploratory analysis in the PMBB, we performed additional single-variant association analyses for each of the six significant variant-phenotype associations in the UKB. Direct replication using the original significant four- or five-digit International Classification of Diseases Ninth Revision (ICD-9) code from the PMBB was tested for each variant-phenotype association. Where there were insufficient case numbers in the UKB, we used the broader three-digit ICD-9 code. Out of six novel associations reaching FDR < 0.1, one (*rs140799351*) showed P < 0.05 in the UKB at the five-digit ICD code level, reaching study-wide significance (Table 1 and Supplementary Table 4). For the remaining putative novel associations, the VPS53 uORF stop-strengthening variant did not replicate, although the direction of effect is consistent with results from the PMBB. Finally variants in SHMT2 could not be replicated because there were fewer than 20 cases in the UKB cohort, and MOAP1 could not be replicated because this variant was absent from the UKB.

**Disease-associated uORF variants change protein expression.** To elucidate the possible biological consequences of UTC and stop-strengthening variants, we selected three PheWAS association signals in the discovery analysis for functional assessment. To determine if these variants could affect protein expression, we measured the expression of a set of dual-luciferase reporters in HEK293T cells for PMVK, VPS53, and the BCL2L13 uORF variants. We compared the expression of the wild-type 5′UTR sequence for PMVK, VPS53, and BCL2L13 cloned upstream of a Firefly Luciferase ORF to two variant sequences—one with the predicted uORF start codon removed, and a second sequence with the PheWAS-significant stop-strengthening variant inserted. For VPS53, we also tested the effect of a mutation changing a tryptophan UGG codon to a UAG UTC (Fig. 5b). Across all

constructs, we observed a significant reduction in expression of the downstream ORF when the PheWAS-significant stop-strengthening variant was introduced (Fig. 5). Introducing a new UTC in the 5′UTR of VPS53 also significantly reduced reporter protein expression relative to the wild-type sequence. Similar results were obtained from assays performed in HeLa cells (Supplementary Fig. 12).

In all the tested constructs, UTC and stop-strengthening variants decreased relative Firefly expression. These data are consistent with the hypothesis that UTC or stop-strengthening variants are under negative selection because they decrease the probability of translation initiation at downstream coding sequences. These results are congruous with our genetic analysis, and imply that UTC and stop-strengthening variants represent a new class of functional variation in 5′UTRs capable of causing loss-of-function of downstream coding genes.

**Replication of novel associations by loss-of-function gene-burden studies.** Results from reporter-gene experiments showed that UTC and stop-strengthening variants could decrease expression of the downstream protein for PMVK, VPS53, and BCL2L13. Our findings implied that uORF UTC and stop-strengthening variants cause phenotypic consequences through potential loss-of-function of the downstream protein-coding gene. To further validate this hypothesis, we performed a gene burden test by aggregating rare predicted loss-of-function (pLOF) protein-coding variants in the PMBB and UKB for each significant uORF-PheWAS association. Rare (MAF ≤ 0.1%) pLOF variants were defined as frameshift insertions/deletions, gain/loss of stop codon, or disruption of canonical splice site dinucleotides. Predicted deleterious rare (MAF ≤ 0.1%) missense variants were defined as those with Rare Exonic Variant Ensemble Learner (REVEL)[40] scores ≥0.5 and included in the set of pLOF protein-coding variants for gene-burden analyses. These studies could confirm that pLOF in the protein CDS of the uORF-regulated gene causes the same phenotype as the uORF UTC or stop-strengthening variants. Indeed, similar loss-of-function gene burden approaches using rare protein-coding variants have successfully been applied to

**Table 1 Significant Novel Associations in PheWAS of Penn Medicine BioBank.**

| Variant (Gene SNP, uORF effect) | Novel associations (Phenotype (Phecode)) | OR (95% CI) | P value | Cases | Controls | Replication | | |
|---|---|---|---|---|---|---|---|---|
| | | | | | | UKB | PMBB LOF | UKBB LOF |
| PMVK* (rs181302437) UAG > UAA | 250.13 (T1D - ophthalmic manifestations) | 27.29 (6.88-108.29) | 2.58E-06 | 23 | 5189 | No | No | Yes (250.13, P = 7.27e-03) |
| | 250.14 (T1D - neurological manifestations) | 22.71 (5.86-87.97) | 6.20E-06 | 25 | 5189 | No | No | No |
| VPSS3 (rs35915949) UGA > UAA | 250.22 (T2D - renal manifestations) | 7.79 (2.87-21.17) | 5.73E-05 | 136 | 5189 | No | No | No |
| | 300.10 (Anxiety disorder) | 0.64 (0.53-0.77) | 4.23E-06 | 1060 | 6939 | No | No | No |
| NALCN (rs139848407) CAA > UAA | 300.00 (Anxiety disorders) | 0.69 (0.58-0.82) | 2.00E-05 | 1249 | 6939 | No | No | No |
| | 270.33 (Amyloidosis) | 38.92 (7.49-202.36) | 1.34E-05 | 30 | 7727 | No | No | Yes (270.00, P = 0.0264) |
| BCL2L13† (rs140799351) UGA > UAA | 610.00 (Benign mammary dysplasias) | 270.57 (19.69-3718.08) | 2.80E-05 | 55 | 7689 | No | Insufficient variants | Insufficient variants |
| | 187.20 (Malignant neoplasm of the testes) | 331.41 (21.68-5065.35) | 3.03E-05 | 26 | 7700 | Yes (187.20, P = 2.09e-4) | Insufficient variants | Insufficient variants |
| | 187.00 (Cancer of other male genital organs) | 220.01 (15.67-3089.83) | 6.31E-05 | 34 | 7700 | Yes (187.00, P = 3.33e-4) | Insufficient variants | Insufficient variants |
| SHMT2 (rs28365863) UAG > UAA | 527.00 (Diseases of the salivary glands) | 6.37 (2.60-15.65) | 5.27E-05 | 90 | 9774 | Insufficient cases | Yes (527.00, P = 5.515e-03) | Insufficient cases |
| MOAP1 (rs116450723) UAC > UAA | 350.00 (Abnormal movement) | 4.99 (2.20-11.33) | 1.22E-04 | 362 | 9414 | No (variant not present in UKB) | No | No (variant not present in UKB) |

*As of the Gencode 32 release the 5′ UTR PMVK annotation (September 2019) was shortened to exclude this uORF; however inspection of the raw ribosome profiling reads from Ji et al.[16] in conjunction with nearby transcription start sites annotated in FANTOM5 confirm the presence of a longer PMVK 5′UTR isoform (Supplementary Fig. 7).
†The stop-strengthening variant in BCL2L13 affects a minor transcript isoform, and is also annotated as a synonymous mutation on the primary BCL2L13 transcript.

identify both known and new gene-disease associations in the PMBB and UKB[37,41].

Of six PheWAS-significant associations uncovered in our discovery analysis (FDR < 0.1), two associations were replicated by an independent loss-of-function gene burden test in either the UKB or PMBB. The associations between *PMVK* and diabetes, and *SHMT2* and diseases of the salivary gland, were replicated in the UKB and PMBB respectively (*PMVK* P = 0.00727, *SHMT2* P = 0.005515, Table 1 and Supplementary Table 4). Although no significant LOF-burden association for PMVK was replicated in the PMBB, pLOF of *PMVK* was nominally associated with impaired fasting glucose (P = 0.0235). A second uORF-disease association was replicated for *NALCN* and the parent three-digit parent PheCode of disorders of plasma protein metabolism in the UKB (P = 0.0264). Gene-disease associations for *BCL2L13* could not be replicated in either the PMBB or UKB due to lack of carriers for pLOF variants. Ultimately this analysis confirmed that loss-of-function gene burden tests using protein-coding variants are associated with the same phenotype for two uORF stop-strengthening variants. This evidence of allelic heterogeneity for these phenotypes further strengthens the likelihood that uORF stop-strengthening variants can cause loss-of-function of downstream protein-coding genes.

## Discussion

By combining large databases of human genetic variation with ribosome profiling, we identified two categories of variants in 5′ UTRs capable of causing reduced translation of downstream coding genes. These variants either introduce UTCs in uORFs or strengthen uORF stop sites. Given that ~50% of human protein-coding genes are estimated to be under translational control by uORFs, these findings provide a framework for interpreting the functional significance of 5′UTR variation for a large fraction of human genes.

Using these variants, we additionally identified gene-disease associations in the PMBB and replicated one of these associations in independent single-variant association tests in the UKB. Two associations involving stop-strengthening variants in *PMVK* and *SHMT2* and one involving a UTC in *NALCN* were also replicated using protein-coding mutations in loss-of-function gene burden tests. These results provide independent validation of uORF variant-phenotype associations uncovered through the PMBB discovery analysis and demonstrate that uORF stop-strengthening and UTC variants associate with the same phenotype as pLOF coding mutations in downstream coding sequences. In support of these conclusions, we have shown that introducing UTCs and stop-strengthening variants in translated uORFs decreases protein expression of downstream genes in reporter assays. These findings establish that uORF UTC and stop-strengthening variants can have functional consequences on protein expression and are associated with disease in humans. Finally, of the 4392 genes with translated uORFs used for this analysis, 1121 (26%) are also annotated as having pathogenic CDS variants in ClinVar, suggesting that UTC and stop-strengthening variants in these genes may have additional utility for the diagnosis of rare disease.

Our results suggest uORF translation has broad roles in regulating CDS translation. Translation initiation is rate-limiting for post-transcriptional protein production and selection against variants disrupting translation elongation (UTCs) or termination at uORFs (stop-strengthening variants) may reflect the importance of preserving translation initiation efficiency at the CDS. This suggested mode of regulation is in-line with observations that *cis*-regulatory relationships between uORFs and downstream coding sequences are frequently conserved across vertebrates

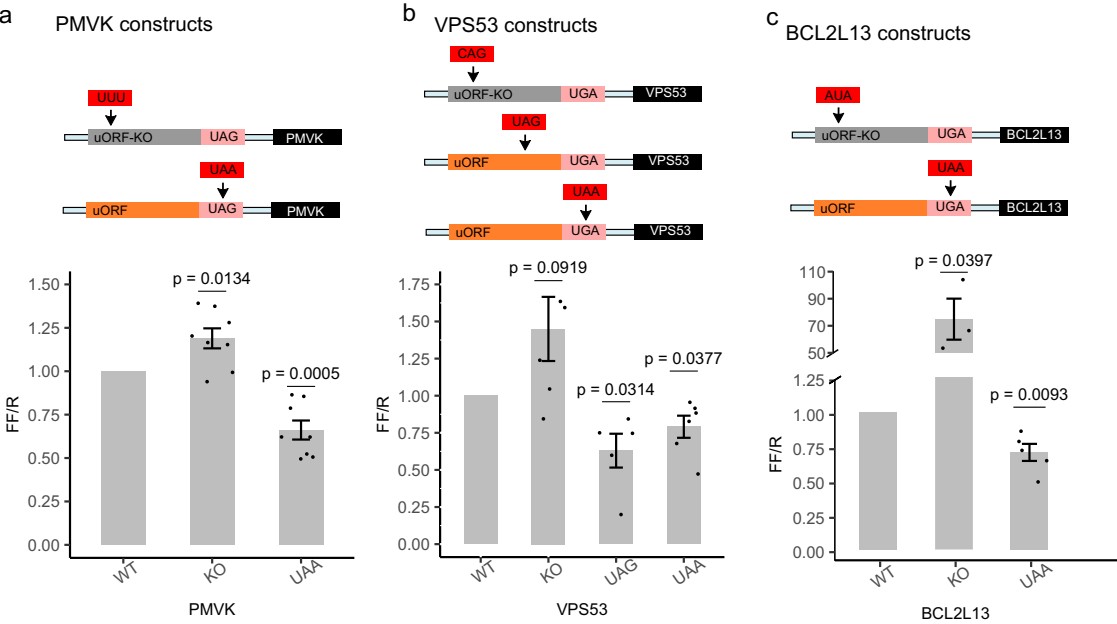

**Fig. 5 Reporter gene assays for translated uORF stop-introducing and stop-strengthening variants.** Dual-luciferase reporter assay quantifies relative expression for uORFs with UTC and stop-strengthening variants associated with EHR phenotypes by PheWAS. Experimental 5′UTRs for **a** *PMVK*, **b** *VPS53*, and **c** BCL2L13 for uORF KO, stop-strengthened, or stop-introduced variants are shown. Bars represent co-transfected Firefly to Renilla Luciferase luminescence ratios normalized to associated wild-type 5′UTRs in HEK293T cells measured 48 h post-transfection. P values from one-sample *t*-test for each condition are displayed above each column. Error bars represent mean + SEM of at least three independent experiments. Raw luciferase measurements are included in the Source Data file.

while features conferring strong uORF repression are less maintained[35,42]. For stop-strengthening variants, the increased translation termination efficiency could accelerate ribosomal release from the mRNA transcript, thus decreasing downstream CDS translation. This mechanism is consistent with previous data in human cell lines showing that broadly decreasing translation termination efficiency by global knockdown of eRF3A increases translation of genes under uORF-repression[43], and that stronger UAA termination signals are less likely to induce translation termination pauses in yeast[23]. For UTC variants, the introduction of stop codons in the uORF can lead to ribosome stalling and subsequent collisions that further repress CDS expression[44,45]. This early translation termination in uORFs might also facilitate greater rates of premature ribosome release from the mRNA transcript, or activate nonsense mediated decay (NMD). The former mechanism is supported by a previously published example of a UTC variant found to decrease ribosome occupancy without affecting mRNA expression[46]. While a handful of translated uORFs that activate NMD have been described in the literature[47–49], whether uORF-activated NMD broadly regulates protein expression remains an open question. Indeed, depletion of UPF1, a central component of the canonical NMD pathway, produced only minimal changes in uORF-containing mRNAs abundance in human cell lines[43].

The capacity for translated uORFs to produce functional micropeptides independent of regulating CDS expression remains an area of active investigation. In canonical protein-coding regions of the genome, amino acid substitutions in critical protein domains can be highly deleterious for cellular functioning and fitness. Previous studies have found that uORF-encoded peptides show evidence of amino acid conservation using statistical tests relying on a null hypothesis of neutral selection[16]. It is unclear if the conclusions drawn from these approaches account for the possibility that codon-optimality constrains variation within uORFs rather than amino acid identity. In contrast, we do not observe similar constraints on missense mutations in translated uORFs, suggesting that amino acid substitutions within most uORF-encoded micropeptides are well-tolerated in humans. This was also the case for other non-canonical translated ORFs, including 3′UTR ORFs, pseudogenes, and lncRNAs, that are not thought to widely encode for functional micropeptides. Although a handful of functional micropeptides have been identified previously, our analysis implies that most ncORFs do not produce peptide products whose function depends on their amino acid composition. It is also important to note that ribosomes are among the most abundant proteins within cells, occupying ~5% of the entire intracellular volume[50]. As improvements in ribosome profiling facilitate deeper characterization of the translatome, observations of widespread translation in ncORFs should be interpreted cautiously in light of potential functionality.

Of the variant-phenotype associations uncovered by our study, only the association between *BCL2L13* and cancer has been previously reported[51,52]. Interestingly, bi-allelic loss-of-function mutations in *SHMT2* have recently been described in a novel brain and heart developmental syndrome involving spastic paraparesis and ataxias[53]. Indeed, in addition to the phenome-wide significant association with diseases of the salivary gland uncovered in our study, the SHMT2 uORF stop-strengthening variant was nominally associated with several Phecodes related to cardiac and movement disorders in the PMBB (Supplementary Table 6), including congenital anomalies of the great vessels (ICD 747.13, $P = 0.0117$), abnormal involuntary movements (350.1, $P = 0.0238$), abnormality of gait (350.2, $P = 0.02575$), Mobitz II AV block (426.22, $P = 0.03432$), and Arrhythmia (cardiac) NOS (427.5, $P = 0.04977$). These additional nominal associations suggest that *SHMT2* uORF variants may be capable of contributing to similar phenotypic consequences as described in loss-of-function mutation carriers, however further studies are needed to investigate this possibility. Finally, the novel association between stop-strengthening and pLOF variants in *PMVK* with

diabetes further strengthens existing genetic and epidemiological evidence linking the mevalonate pathway to diabetes. *PMVK* encodes for phosphomevalonate kinase, an enzyme in the mevalonate pathway catalyzing the conversion of mevalonate-5-phosphate to mevalonate-pyrophosphate downstream of HMG-CoA reductase. Multiple randomized clinical trials have shown that inhibiting HMG-CoA reductase with statins increases the risk of developing new-onset type 2 diabetes in a dose-dependent manner, although the mechanism driving this association has remained elusive[54–56]. Moreover, genetic variants in and near the *HMGCR* gene that are associated with lowered LDL cholesterol levels have been similarly shown to confer an increased risk of developing diabetes[57,58], suggesting that decreased *HMGCR* activity contributes to diabetes pathogenesis. Our data is the first to establish a putative link between *PMVK* and diabetes. Given the shared involvement of *PMVK* and *HMGCR* genes in the mevalonate pathway, it is possible that variants in both these genes confer an increased risk of diabetes through a similar mechanism, however additional studies will be needed to further elucidate the precise relationship between *PMVK* and diabetes.

A limitation of our analysis is that we cannot directly assess the impact of additional factors on uORF-mediated translational regulation which may change the impact of UTC or stop-strengthening variants on protein expression. As an example, a pathogenic mutation 279 base pairs upstream of the BMPR2 CDS was previously found to cause an unexplained ~93% decrease in protein expression in a patient with idiopathic pulmonary arterial hypertension[59]. We note that this variant introduces a UGA stop codon which interrupts two overlapping UUG-intiated uORFs mapped by ribosome profiling beginning 290 and 305 base pairs upstream of the BMPR2 CDS respectively corroborating our findings that UTC variants are capable of decreasing downstream protein expression. In contrast, pathogenic UTC variant in the *U2HR* gene has previously been reported to confer gain-of-function in Marie Unna hereditary hypotrichosis[60]. However, missense variants in this uORF also confer gain-of-function effects, suggesting that these mutations might contribute to pathology through disrupting alternative regulatory mechanisms or a functional micropeptide. Indeed, previous studies have shown that a multitude of factors may impact uORF regulatory function, and it is likely that in addition to decreasing protein expression, UTC variants are also capable of causing gain-of-function in some uORF-regulated genes. Dissecting these effects remains a challenge for future studies.

Finally, we note that being a hospital-based biobank, participants in the PMBB are generally less healthy than the general population. As phenotypes within broader disease Phecode families are often highly correlated, we sought to replicate associations uncovered in the discovery analysis by first testing for a specific hypothesis-driven phenotype association in addition to related phenotypes in the corresponding Phecode families. We recognize that controlling for type 1 error in this framework remains challenging, however to remedy this we sought additional confidence by further replicating significant uORF-variant associations through loss-of-function gene-burden analyses. Moreover, the relative enrichment in diseased individuals in the PMBB may account for why few associations discovered in our analysis of the PMBB are replicated in the UKB which contains a healthy volunteer selection bias[61]. Indeed we were unable to test for an association for two of the six PMBB associations due to an inadequate number of individuals having the phenotype in UKB. As hospital-based biobanks become more prevalent these unreplicated associations should be revisited and confirmed.

Understanding and interpreting the impact of noncoding genetic variation is a fundamental challenge in biology. Many mutations affecting uORFs are known to cause disease[62–65], but until now, most studies have focused on variants which destroy start codons, stop codons of existing uORFs, or those that create new inhibitory uORFs. By examining patterns of genetic variation within translated uORFs, we have uncovered two categories of variation affecting 5′UTRs that may lead to loss-of-function in associated genes. We have used these variants to identify gene-disease associations, and provide evidence for their ability to impact downstream gene expression. Our approach demonstrates the power of integrating population-scale databases of human genetic variation with cellular-scale -omics data to identify patterns of how variation impacts regulatory elements. Taken together, our data broadens the scope of functional translational regulation by uORFs in the transcriptome and expands approaches for interpreting functional genetic variation in 5′UTRs.

## Methods

**Annotation of translated non-canonical open reading frames.** Non-canonical ORF (ncORF) annotations encompassing 5′UTR ORFs (uORFs), 3′UTR ORFs (dORFs), long-noncoding RNA ORFs (lncRNA), and pseudogene ORFs were retrieved from Ji et al.[16]. These ncORFs were mapped by ribosome-profiling in human BJ fibroblasts and MCF10A breast epithelial cells using the RibORF algorithm. Using the final set of genomic coordinates for ncORFs identified in this study, we converted these coordinates to match hg38 annotations using the UCSC LiftOver executable (https://genome.ucsc.edu/cgi-bin/hgLiftOver). Out of 10,007 distinct non-canonical uORFs mapped in the original study, 27 whose length changed after conversion (N = 5 uORFs, 4 dORFs, 16 lncRNA ORFs, 2 pseudogenes) were excluded from subsequent analyses. Each Refseq mRNA ID for each ORF-associated RNA transcript was annotated to its associated Ensembl transcript ID using the BioMart database v86 annotations. The first three nucleotides of each ORF were used as start codons for downstream analyses. The final three nucleotides of each ORF were used as stop codons for downstream analyses. 5′ and 3′ UTR definitions used in this study are derived from the Ensembl v86 annotations.

**Quality filtering and annotation of variants from gnomAD version 3.** Variants from gnomAD 3 release were downloaded from the gnomAD browser website (https://gnomad.broadinstitute.org/downloads). A set of high-confidence variants were obtained by removing those failing the Filter column (Filter! = PASS) from the gnomAD version 3 vcf files using bcftools (version 1.9), and those falling in low complexity regions (lcr! = 1). This set of variants was used for all downstream analyses. We additionally removed variants where the total observed allele number was at least than 80% of the maximum number of sequenced alleles to control for differences in sequencing depth in the gnomAD WGS dataset. The remaining set of high-confidence variants was overlapped with genomic coordinates for annotated ncORFs, 5′UTR sequences, and annotated protein-coding sequences using bedtools (version 2.27.1) intersect with the -u and -b flags. The predicted consequence of each variant was obtained using the Ensembl Variant Effect Predictor (VEP, version 98.2) based on hg38 gene models obtained from Ensembl. VEP consequences were further filtered to only include the predicted consequence for the canonical Ensembl transcript as determined in[12].

**Positional constraint analysis using variants from gnomAD.** For the positional constraint analysis we applied the MAPS metric to each variant set. We developed a MAPS model following previous methods[12]. The set of synonymous protein-coding variants are used as a baseline measurement for neutral selection, and the proportion of singletons in a variant class are adjusted for differences in mutation rates due to local sequence context[11,12]. We trained our model by regressing the observed proportion of singleton-synonymous variants for each trinucleotide context within protein-coding regions of the genome using previously published context-dependent mutation rates derived from intergenic noncoding regions of the genome[12]. Since negative selection prevents deleterious mutations from becoming common in human populations, more deleterious mutations—including those disrupting essential splice sites or introducing premature termination codons—are also more enriched for singletons compared to neutral variants.

MAPS scores for a given set of variants are calculated as follows:[9,11,12]. for a given set of variants, we use the MAPS model to determine the expected number of singletons that should be observed, based on the transformed mutation rates which account for trinucleotide context and methylation levels. To calculate the MAPS score, we take the observed number of singletons for this set of variants, and subtract the expected number of singletons calculated using the MAPS model. We then divide this value by the number of variants total to obtain the proportion of singleton variants adjusted for mutation context.

To estimate of MAPS scores for missense-causing mutations in canonical protein-coding sequences within the genome, we selected the subset of SNVs in gnomAD with an annotated VEP consequence of missense, and removed SNVs from this set of variants if they had additional VEP annotations that could be considered pLoF. The set of variants used to calculate MAPS scores for pLoF

variants relied on aggregating variants with a VEP annotation of transcript_ablation, splice_acceptor_variant, splice_donor_variant, stop_gained, frameshift_variant, stop_lost, and start_lost terms. The set of synonymous variants used to train the MAPS model was filtered to remove variants with any of the previous predicted high impact annotations, and those with a possible missense consequence.

We computed MAPS scores for each set of variants based on uORF annotations, or 5′UTR annotations from Gencode (GRCh38.p13; https://www.gencodegenes.org/human/release_32.html). Using the set of filtered variants we matched them to uORF positions annotated by their relative position within the uORF reading frame, strand, and codon. We determined how the mutation affected the codon within the translated uORF sequence, and annotated each variant with its consequence on the encoded amino acid. We used these annotations to select variants that could introduce new stop codons (UTC-introducing variants) and those that strengthened existing stop codons within uORFs. For UAA-introducing variants we selected any variant that produced an in-frame UAA stop codon. For each set of stop-introducing or stop-strengthening variants, we selected a set of uORF variants matching the underlying trinucleotide context of each experimental set of variants. MAPS scores for these variant sets were computed and confidence intervals were determined by resampling from each variant set with replacement over 10,000 iterations.

For codon optimality analysis, we used the set of codon stability coefficients (CSC) scores derived from SLAM-seq in 562 cells obtained from 10.7554/eLife.45396.006[34]. Optimality decreasing variants were defined as any variant which decreased the CSC score for the encoded codon, and optimality increasing variants were defined as any variant which increased the CSC score for the encoded codon.

Confidence intervals for MAPS scores were calculated using bootstrapping as described[9]. For each set of $n$ variants used to compute a MAPS score, we select $n$ variants randomly with replacement and recalculate MAPS scores. This is repeated over 10,000 permutations and the 5th and 95th percentiles of the MAPS scores distribution are used as confidence intervals. $P$ values for differences in MAPS scores were determined by calculating the proportion of bootstrapped MAPS scores from an experimental group of variants that were larger than those from the control group[9].

**Determining the distribution of stop codons used by uORFs.** Stop codons from each uORF were extracted based on genomic coordinates and the uORF reading frame. Confidence intervals were determined by sampling with replacement from the set of uORF stop codons over 10,000 iterations. For 5′UTR sequences, all stop-codon matching trinucleotides (UGA, UAG, UAA) were extracted from annotated canonical 5′UTR sequences of protein-coding genes in the BioMart Ensembl database (version 86). The set of canonical transcripts annotated in the gnomAD flagship release paper were used to define 5′UTR sequences for this analysis[12]. For each iteration, one stop codon was randomly selected from each 5′UTR and the proportion of UGA, UAG, and UAA trinucleotides selected from all 5′UTR sequences were calculated. This procedure was repeated 10,000 times to form a distribution of UGA, UAG, and UAA trinucleotides in all 5′UTR sequences. This procedure was also repeated for uORF-matched UTR sequence segments that did not overlap known translated uORFs. $P$ values for the depletion of UAA stop codons used in translated uORFs were calculated by determining the number of bootstrap iterations where the frequency of UAA codons from uORFs was higher compared to non-uORF sequences. $P$ values for enrichment of UGA and UAG sequences were calculated by determining the fraction of sampled iterations where fewer UGA and UAG sequences were selected from uORF stop codons compared to all 5′UTRs and uORF-matched 5′UTR sequences respectively.

**Assessing variant conservation using genome-wide phyloP scores.** PhyloP scores for each base were downloaded from the UCSC genome browser (http://hgdownload.cse.ucsc.edu/goldenpath/hg38/phyloP100way/). 1-indexed bigwig files were converted to bed file format using the wig2bed tool from bedops (version 2.4.36; https://bedops.readthedocs.io/en/latest/index.html). These base-level annotations were matched to each uORF base and used to determine the proportion of bases that were significantly conserved (proportion of bases with phyloP score >2). Possible inframe stop-codon creating positions were identified based on mapped reading frames for each uORF. These sites were extracted and further categorized by whether a SNV could create a UGA, UAG, or UAA stop codon. Some positions could be mutated to either a UGA, UAG, or UAA codon and these were considered separately from potential UGA, UAG, or UAA-creating positions. We have included all potential stop-introducing positions as Supplementary Data 1. As a control we used phyloP scores for genomic positions with the potential to create non-uORF UGA, UAG, or UAA trinucleotides by mutation, but matched by distance to CDS in 10-base pair windows.

Start-disrupting genomic positions were annotated as those mutating the second or third position in the first codon of each translated uORF. Conservation based on phyloP scores were assessed for start-disrupting positions similar to potential stop-introducing positions. As a control we compared phyloP scores for uORF start-disrupting positions to out-of-frame start-disrupting positions within annotated uORFs, and a set of NTG start-disrupting variants that were not part of

translated uORFs but matched by distance to the CDS as determined by 10-bp windows.

$P$ values were determined by sampling with replacement from each set of variants 10,000 times and re-calculating the proportion of significantly conserved bases (phyloP score >2). The distribution of the fraction of conserved base positions were then compared against different sets of variants, and the $P$ value was defined as the fraction of samples where one group was higher than the other.

**Setting and study participants.** All individuals who were recruited for the PMBB are patients of clinical practice sites of the University of Pennsylvania Health System. Appropriate consent was obtained from each participant regarding storage of biological specimens, genetic sequencing, access to all available electronic health record (EHR) data, and permission to recontact for future studies. The study was approved by the Institutional Review Board of the University of Pennsylvania and complied with the principles set out in the Declaration of Helsinki. Replication analyses were conducted using the whole exome sequencing (WES) dataset from the UKB.

**Genetic sequencing.** This PMBB study dataset included a subset of 11,451 individuals in the PMBB who have undergone WES. For each individual, we extracted DNA from stored buffy coats and then obtained exome sequences generated by the Regeneron Genetics Center (Tarrytown, NY). These sequences were mapped to GRCh37 as previously described[37]. Furthermore, for subsequent phenotypic analyses, we removed samples with low exome sequencing coverage (i.e., less than 75% of targeted bases achieving 20x coverage), high missingness (i.e., greater than 5% of targeted bases), high heterozygosity, dissimilar reported and genetically determined sex, genetic evidence of sample duplication, and cryptic relatedness (i.e., closer than third degree relatives), leading to a total of 10,900 individuals.

For replication studies in UKB, we interrogated the 32,268 individuals of European ancestry (based on UKB's reported genetic ancestry grouping) with International Classification of Diseases Tenth Revision (ICD-10) diagnosis codes available among the 49,960 individuals who had WES data as generated by the functional equivalence (FE) pipeline. We focused our replication efforts on 32,268 individuals after removing samples with poor genotype quality, individuals closer than third degree relatives, and those with dissimilar reported and genetically determined sex. The PLINK files for exome sequencing provided by UKB were based on mappings to GRCh38. Access to the UKB for this project was from Application 32133.

**Variant annotation and selection for association testing.** For both PMBB and UKB, genetic variants were annotated using ANNOVAR[66] as 5′ untranslated region (5′ UTR), pLOF, or missense variants according to the NCBI Reference Sequence (RefSeq) database[66,67]. Rare (MAF ≤ 0.1%) pLOF variants were defined as frameshift insertions/deletions, gain/loss of stop codon, or disruption of canonical splice site dinucleotides. Predicted deleterious rare (MAF ≤ 0.1%) missense variants were defined as those with REVEL[40] scores ≥0.5. pLOF and REVEL-informed missense variants were selected for gene burden testing to validate the robustness of significant uORF variants' corresponding gene-disease associations.

**Clinical data collection.** ICD-9 and ICD-10) disease diagnosis codes and procedural billing codes, medications, and clinical imaging and laboratory measurements were extracted from the patients' EHR for PMBB. ICD-10 encounter diagnoses were mapped to ICD-9 via the Center for Medicare and Medicaid Services 2017 General Equivalency Mappings (https://www.cms.gov/Medicare/Coding/ICD10/2017-ICD-10-CM-and-GEMs.html) and manual curation. Phenotypes for each individual were then determined by mapping ICD-9 codes to distinct disease entities (i.e., Phecodes) via Phecode Map 1.2 using the R package "PheWAS"[38,68]. Patients were determined to have a certain disease phenotype if they had the corresponding ICD diagnosis on two or more dates, while phenotypic controls consisted of individuals who never had the ICD code. Individuals with an ICD diagnosis on only one date as well as individuals under control exclusion criteria based on PheWAS phenotype mapping protocols were not considered in statistical analyses.

For UKB, we used the provided ICD-10 disease diagnosis codes for replication studies, and individuals were determined to have a certain disease phenotype if they had one or more encounters for the corresponding ICD diagnosis given the lack of individuals with more than two encounters per diagnosis, while phenotypic controls consisted of individuals who never had the ICD code. Individuals under control exclusion criteria based on PheWAS phenotype mapping protocols were not considered in statistical analyses.

**Association studies.** A PheWAS approach was used to determine the phenotypes associated with 5′ UTR variants predicted to create new UAA UTCs, or strengthen existing uORF stop sites and carried by individuals in PMBB for the discovery experiment[38]. Each disease phenotype was tested for association with each uORF variant using a logistic regression model adjusted for age, age[2], sex, and the first ten principal components (PCs) of genetic ancestry. We used an additive genetic model to collapse variants per gene via an extension of the fixed threshold approach[69]. Given the high percentage of individuals of African ancestry present in the

discovery PMBB cohort, association analyses were performed separately in European ($N = 8198$) and African ($N = 2172$) genetic ancestries and combined with inverse variance weighted meta-analysis. Only 5′ UTR variants with at least five total alternate alleles in PMBB were selected for univariate PheWAS analyses in the discovery phase while variants with greater than half of the genotypes annotated as missing due to low quality were excluded. This resulted in a final set of $N = 10$ variants. Our association analyses considered only disease phenotypes with at least 20 cases, leading to the interrogation of 800 total Phecodes. All association analyses were completed using R version 3.3.1 (Vienna, Austria).

We evaluated the robustness of significant uORF-phenotype associations in the same PMBB discovery cohort by aggregating pLOF and predicted deleterious missense variants in each uORF's corresponding gene into a "gene burden" for hypothesis-driven association with the significant phenotype from discovery. Only gene burdens with at least five total alternate alleles in PMBB were selected for replication studies. All gene burden association studies in PMBB were based on a logistic regression model adjusted for age, age², sex, and the first ten PCs of genetic ancestry.

Additionally, we replicated our findings in UKB for significant uORF associations in the PMBB discovery using (1) hypothesis-driven univariate association studies for the same uORF variants and (2) hypothesis-driven gene burden collapsing pLOF and predicted missense variants for the corresponding genes. Only uORF variants and gene burdens with at least five total alternate alleles in PMBB were selected for replication studies. Association statistics were calculated similarly to PMBB, such that each disease phenotype was tested for association with each gene burden or single variant using a logistic regression model adjusted for age, age², sex, and the first ten PCs of genetic ancestry. Replication significance was defined using a $P$ value threshold of 0.05. All association analyses for PMBB and UKB completed using R version 3.6.1.

**Construction of expression vectors.** The test plasmids used a modified pGL4.12 [luc2CP] (Promega) vector backbone where the control of expression of the Firefly ORF was modified by the addition of an upstream CMV promoter. The modified pGL4.12 vector was linearized using Bgl-II and MreI restriction sites. Hybrid 5′ UTR fragments containing the entire 5′UTR sequence and the first 91 nucleotides of the Luc2 Firefly ORF were produced by gBlock synthesis and received from Integrated DNA Technologies using sequences in Supplementary Table 5. Test plasmids were constructed by sub-cloning these hybrid 5′UTR sequences for PMVK, VPS53, and BCL2L13 into the modified pGL4.12 vector to preserve the uORF-CDS relationship for each construct. Correct fragment insertion was verified for each engineered construct by sanger sequencing. For PMVK and BCL2L13, the entire annotated 5′UTR sequence was used. For VPS53, because of a G-rich sequence in the 5′UTR upstream of the uORF complicated synthesis of the gene's entire 5′UTR fragment, we removed the first 75 nucleotides of the annotated 5′ UTR sequence. Construct assembly was accomplished using the NEB Hi-Fi assembly protocol following the manufacturer's instructions.

**Cell culture and transfections.** HEK293T cells were used for conditional expression of reporter genes. For transient transfections, HEK293T cells were split 1 day before transfection and seeded in 24-well plates at a density of 100,000 cells per well. Two micrograms of the test Firefly reporter plasmid was transfected into each well using Lipofectamine 3000 following the manufacturer's protocol using 1.5 uL of transfection reagent and 0.5 uL of the P3000 reagent for each well. As a control for transfection efficiency, 0.02 ug of the pRL-CMV Renilla Luciferase plasmid (Promega Accession No. AF025843) was co-transfected with firefly luciferase plasmids. Biological replicates were obtained by transfecting cells from separate passages on separate days using newly prepared reagents. All transfections were repeated using the HeLa cell line. Dulbecco's modified Eagle's medium (DMEM) supplemented with 10% (v/v) fetal bovine serum and antibiotics was used for all cell culture.

**Luminometry assays.** Luminescence was measured using the Promega Dual-Luciferase Reporter Assay System (E1910) following the manufacturer's protocol. Cells were lysed by adding 100 uL of lysis buffer; 10 uL of each lysate was transferred to a black opaque 96-well plate. The ratio of Firefly to Renilla luminescence with a microplate reader by automatic injection of the Luciferase Assay Reagent II and Stop & Glo reagents. Biological replicates were obtained by transfecting cells from separate passages on separate days using newly prepared reagents. Luminescence measurements were compared within each set of transfections and statistical significance was determined using a one-sided $T$ test comparing the firefly to renilla expression ratio of each test construct normalized to the wild-type construct.

**Reporting Summary.** Further information on research design is available in the Nature Research Reporting Summary linked to this article.

## Data availability

The set of gnomAD variants obtained from 71,702 whole genome sequences used for the MAPS analysis are available from https://gnomad.broadinstitute.org/downloads. The set

of mapped ncORFs from ribosome profiling studies used for the analyses presented are available from 10.7554/eLife.08890.023. This set includes 5′UTR (uORF), 3′UTR (dORF), long-noncoding RNA, and pseudogene ORFs mapped by the RibORF algorithm. Codon stability coefficient (CSC) scores used in the analyses were downloaded from 10.7554/eLife.45396.006. Supplementary Data 1 is provided as a text file accessible at 10.5281/zenodo.4536050[70]. Individual-level data from the PMBB are not publicly available due to research participant privacy concerns; however, requests from accredited researchers for access to individual-level data relevant to this study can be made by contacting the corresponding author. Up-to-date summary data for genetic variants captured using WES in the PMBB can be accessed via the PMBB Genome Browser (https://pmbb.med.upenn.edu/allele-frequency/). Base-level conservation phyloP values were obtained from the UCSC Genome Browser at the following link: https://genome.ucsc.edu/cgi-bin/hgTrackUi?db=hg38&g=cons100way. Source data are provided with this paper.

## Code availability

All scripts used in this analysis except for those generating PheWAS results and plots can be accessed at[70].

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

## Acknowledgements

We thank Dr. Benjamin Voight, Dr. Robert Heuckeroth, Matthew Gazzara, and members of the Barash lab for discussions and thoughtful feedback. We gratefully acknowledge the staff of the Regeneron Genetics Center for whole-exome sequencing of PMBB participants. D.L. work was supported by the NIH grant 5T32HG000046-20 and the Blavatnik Family Foundation. Y.B. and D.L. work was supported by R01 GM128096.

## Author contributions

D.S.M.L., L.R.G., and Y.B. conceived and designed the project. D.S.M.L. performed the analyses under the guidance of L.R.G. and Y.B. D.S.M.L. and A.K. performed in vitro experiments. J.P. performed the PheWAS association studies under the guidance of D.J.R. and M.D.R. J.P. wrote the methods for the PheWAS analysis. J.P. A.B., M.D.R., and D.J.R. performed data acquisition for the PheWAS analysis. D.S.M.L. wrote the paper and all authors contributed to editing the paper.

## Competing interests

A.B. is employed by Regeneron Pharmaceuticals. L.R.G. is employed by Janssen Research and Development. The rest of the authors declare no competing interests.
