## [Peer Review File · Nature Communications]

Reviewer #1 (Remarks to the Author):

This study by Lee et al. assesses the impact of variants that disrupt start and stop codons of translated upstream open reading frames using signals of negative selection from >70,000 individuals in gnomAD. The authors identify variants creating stop variants in uORF sequences and that strengthen uORF stop codons that look to be deleterious. Finally, the authors use biobank datasets to look for association of these variants with phenotypes.

My opinions on this paper are a tale of two halves. I think the first half, which uses MAPS scores to assess the deleterious of stop gain and strengthening variants is elegant and interesting. However, the second half of the paper on phenotype associations is weak and unconvincing. I also think the authors make some strong claims (such as a 'loss-of-function' effect) that are not supported by their data. Below are my specific comments.

Major:

1. The assertion that these variants can cause LoF is not supported by the data. Yes, they may often (but perhaps not always) reduce protein translation, however there is no evidence that this will cause a complete LoF (and therefore be useful for clinical variant interpretation). This should therefore not be stated in the title, abstract or conclusions. Specifically:
 - a. Whilst MAPS scores are increased, this does not suggest a mechanism.
 - b. The reporter assays show reduced protein levels, but not with a large effect – none of the tested examples reduce protein levels by more than ~40%.
 - c. The authors do not report example disease-causing variants (as opposed to associations)
 - d. The authors could add support to this hypothesis by looking if MAPS scores are enriched upstream of LoF intolerant genes (low LOEUF scores)
 - e. Looking at UTC gained variants in ClinVar, there seems to be only a single example, and this seems to be associated with up-regulation of protein expression.
2. I find Figure 1b to be very hard to follow
 - a. The 'All 5'UTR' point is for some reason in the 'ncORF' section
 - b. In part ii. The sub-points do not link well to either the legend or the text in the results – I found myself lost as to what most of these points actually represent
 - c. If this is not represented by one of the points, the authors should include UTC creating SNVs in non-uORF parts of the UTR.
3. The result for CDS-overlap removing vs not variants is not at all convincing given the range of possible values (wide CIs) with the small number of variants assessed – both could have MAPS identical to the all UTC point. Furthermore, calculating significance against the all uORF SNVs point does not make sense to argue this point.
4. I am unconvinced on multiple aspects of the association analyses as presented. Specifically:
 - a. The PMVK variant appears to not be part of the UTR for all GRCh38 transcripts on UCSC – have you confirmed that the effect of this variant is indeed on translation and not through reducing RNA levels? Is this variant an eQTL in GTEx?
 - b. BCL2L13 is only a 5'UTR variant in a minor transcript that doesn't appear to be expressed – on the canonical transcript it is synonymous. This is the only one with any convincing replication data.
 - c. The details of the replication analysis are sparse – the numbers of cases and controls should be displayed along with the exact number of tests i.e. how many diabetes related traits were separately tested?
 - d. Two of the replication results for PMVK would not survive any multiple testing and I doubt even the UKBB LoF result for this one would survive if this was done properly. The result from all tests of diabetes phenotypes should be shown.
 - e. The LoF replication analysis is misleading – on reading the methods it seems that missense variants with high REVEL scores were also included. This is not the LoF test that is claimed in the main text and this should be made clear. How many of these associations are driven by missense variants rather than LoF? Was there any filter on variant frequency (i.e. as common pLoFs are likely not true LoF)?
 - f. The authors state "A second uORF-disease association was replicated for NALCN and the parent

PheCode of disorders of plasma protein metabolism in the UKB" – with a p-value of only 0.026 and no multiple-testing correction this claim is untrue.

5. The functional assays would be more convincing if the authors showed the effect of simultaneously removing the uORF start and the stop strengthening variant. Also, why did the authors not assess the effect of the BCL2L13 variant which is the only one with convincing replication data? I imagine this may be because this work was done upstream of, or in tandem with the replication studies.

6. In the discussion the authors claim "These findings establish that uORF UTC and stop-strengthening variants can have functional consequences on gene expression and cause disease in humans" – this is not supported by the data shown. The authors show that these variants may be associated with phenotypes but not that they can cause disease.

7. The analysis in the discussion and presented in the supplementary note includes a lot of assumptions and is hard to assess as included – this should either be included and displayed properly as an analysis in the paper or removed.

Minor:

1. The first sentence of the abstract downplays what is already known about uORFs – there has been plenty of prior research showing the negative role of these elements on translation, but this sentence suggests their function is unknown.

2. The final sentence of the abstract states "translated uORFs are genetically constrained regulatory elements in 40% of human genes" – where does 40% come from? The remainder of the paper uses 50%. Also, this sentence is misleading as it suggests the sequence of the uORFs are constrained, which they show is not the case.

3. The introduction is rather short and a lot of what could be in the introduction is within the results, inflating the size of these sections. For example, there is nothing in the intro about what is currently known about the lack of conservation across uORF sequences. Also, most of the first paragraph of the results would be better placed in the introduction. The introduction also only states that prior work by Whiffin et al. looked at creating new uORFs when they also assessed uORF stop removing variants.

4. How many genes are represented by translated uORFs? How many genes have multiple?

5. How does the enrichment in Figure 1c compare to stop codons used in canonical coding sequences?

6. It would be helpful to see an 'opposite' point to TAA gained UTC – is there any selection against TGA gained UTCs?

7. Conservation was used in Whiffin et al. to assess stop-removing variants not start creating variants – the text on this should be corrected

8. Figure 2c would benefit from an 'all uORF' point

9. It would be informative to reference Schulz et al. Sci Reports 2018 when discussing start removing variants

10. Are stop and start matched variants also matched by UTR position?

11. "Using exome sequencing from the PMBB, we identified heterozygous and homozygous individuals carrying mutations in uORFs which introduce upstream termination codons and stop-strengthening mutations. For the latter class, we focused on variants that introduced TAA stop codons, as the heightened MAPS score for such variants implied these mutations would be most deleterious" – I assume this should say "For the former class"?

12. Table 1 should report variant frequencies not just numbers for cases and controls. There should also be full details on variant numbers for all replication analyses. This should include split data for Europeans and Africans for PMBB.

13. More details should be given on the definition of 5'UTRs

14. The MAPS results for both missense and LoF variants look to be depleted compared to previous work – this may be because the authors deviated from just using the effect on the canonical transcript to, for missense, removing variants with a coding annotation on any transcript, and, for pLoF, including start and stop lost which are not always true LoF.

Reviewer #2 (Remarks to the Author):

This clearly-structured and well-written manuscript from David Lee and colleagues builds upon recent work (most notably from Nicky Whiffin) exploring the degree of purifying selection ("constraint") on 5' UTRs. The rationale for studying these regions is manifold: first translation regulation is a deeply complex and relatively poorly understood process; second, less than half of individuals with rare disease receive a genetic diagnosis and the variants in 5'UTRs are largely ignored owing to a focus on exome sequencing and difficulty prioritizing such variants. Overall, I found this to be a very compelling and string study that focuses on specific classes of 5'UTR variation to which special attention should now be paid in disease studies. I have two concerns that I would like the authors to address.

1. As you know, many genes have multiple isoforms whose 5'UTRs start at different genomic positions. As a result, sequence that is part of a 5'UTR in one isoform could be CDS in another. It was not clear to me how you accounted for the possibility that genetic constraint predicted through higher MAPS scores manifests from purifying selection on CDS in another isoform?

2. Previous work this year from your group showed constraint on G-quadruplexes in 5' UTRs. I was surprised to not see an evaluation of the constraint you observe in this analysis with respect to the presence or absence of a G-quadruplex, as that paper similarly argues that "negative selection acting on central guanines of UTR pG4s is comparable to that of missense variation in protein-coding sequences." To what degree is the constraint you observe on stop-introducing and stop-strengthening mutations in translated uORFs a result, at least in part, on constraint owing to the presence of a G-quadruplex? I am curious because the prior paper makes the argument that such regions are enriched for cis-eQTLs and RBP interactions, which is a completely distinct mechanistic explanation for constraint on these sequences.

Minor

1. The x-axis for Fig3A is unlabelled and thus confusing.

Reviewer #3 (Remarks to the Author):

In this work, the authors nicely describe classes of variants within the 5'UTR that punitively influence translational efficiency by disrupting upstream open reading frames. The authors examine both start and termination codons (and their relative strengths) within these uORFs and examine them from a conservation and evolutionary perspective. They find selective pressure preserving the general structure of the uORF, however the amino acid content was not found to be under the same degree of constraint as missense variants within coding sequence. The authors then select rare variants creating premature stop codons within the uORF of a few genes within large biobank studies to conduct a phenome-wide association study to identify their potential phenotypic consequences. In general this study is well-conceived, constructed, and executed.

My primary concern with the manuscript is the rather confusing way the authors have used "gene expression" and "protein expression" interchangeably in both wording and some aspects of the experimental design. For example, in the first paragraph of the introduction, the authors state "Specific uORFs are known to control gene expression by tuning translation rates of downstream protein-coding sequences". I believe here the authors mean "protein expression". This happens throughout the manuscript, though I believe the primary focus of this work is on protein expression and the translation process. This is further exemplified by the framing of the luciferase reporter assay as determining "if these variants could affect gene expression", when in fact the authors are using this system to directly measure protein product translated from different UTR-based constructs. Clarifying this language will make the focus of the manuscript much more clear and will prevent lots of

confusion.

I will also admit to some disappointment that ribosomal profiling experiments were not attempted, especially given the availability of the luciferase construct. This would have provided additional mechanistic evidence of the role of the uORFs in ribosomal positioning, and would have allowed some investigation of the changes in stop-codon strength and the other factors the authors mention in the limitations section.

A minor concern - the information in table 1 is difficult to read given the column widths - I would convert this to a landscape page.

REVIEWER COMMENTS

Reviewer #1 (Remarks to the Author):

This study by Lee et al. assesses the impact of variants that disrupt start and stop codons of translated upstream open reading frames using signals of negative selection from >70,000 individuals in gnomAD. The authors identify variants creating stop variants in uORF sequences and that strengthen uORF stop codons that look to be deleterious. Finally, the authors use biobank datasets to look for association of these variants with phenotypes.

My opinions on this paper are a tale of two halves. I think the first half, which uses MAPS scores to assess the deleterious of stop gain and strengthening variants is elegant and interesting. However, the second half of the paper on phenotype associations is weak and unconvincing. I also think the authors make some strong claims (such as a 'loss-of-function' effect) that are not supported by their data. Below are my specific comments.

Major:

- 1. The assertion that these variants can cause LoF is not supported by the data. Yes, they may often (but perhaps not always) reduce protein translation, however there is no evidence that this will cause a complete LoF (and therefore be useful for clinical variant interpretation). This should therefore not be stated in the title, abstract or conclusions.*

Specifically:

- a. Whilst MAPS scores are increased, this does not suggest a mechanism.*
- b. The reporter assays show reduced protein levels, but not with a large effect – none of the tested examples reduce protein levels by more than ~40%.*
- c. The authors do not report example disease-causing variants (as opposed to associations)*
- d. The authors could add support to this hypothesis by looking if MAPS scores are enriched upstream of LoF intolerant genes (low LOEUF scores)*
- e. Looking at UTC gained variants in ClinVar, there seems to be only a single example, and this seems to be associated with up-regulation of protein expression.*

We thank the reviewer for their thoughtful and constructive feedback. We appreciate and recognize these concerns and have changed our language throughout the manuscript accordingly in the title, abstract, and conclusions. Importantly we do not wish to overstate our results, rather we hope this work provides a point of entry into further investigations on the impact of uORF-perturbing genetic variants in human disease.

With regards to the reporter assay results we would like to emphasize that use of heterologous reporter gene assays has inherent limitations for quantitative analysis of the effect of single-nucleotide loss-of-function mutations. Gene expression in our luciferase construct is driven by strong viral promoters; our constructs lack the endogenous 3'UTR sequences

associated with each mRNA transcript, and we are using cell lines for which rates of translation and growth tend to be much higher than most cells within the body. Thus, effect sizes observed should be interpreted with caution as they relate to potential *in vivo* effects on endogenous protein expression. As an example, McCabe et al. use the luciferase assay to identify and model protein-coding loss-of-function mutations from patients in the KAL1 gene causing septo-optic dysplasia, finding that these loss-of-function mutations produced a 20-40% decrease in luciferase expression (see Fig. 4C) ¹. We note a similar effect size (20-60% decrease) was observed when loss-of-function 5'UTR variants in the SHOX gene were interrogated by luciferase assay in a separate publication (see Fig. 3) ².

Additionally, as recommended by the reviewer, we have performed an analysis of MAPS scores of uORF variants upstream of the top sextile of LOEUF genes compared to the bottom sextile of LOEUF genes. MAPS scores are nominally higher for uORF UTC / stop-strengthening mutations for the top sextile of LOEUF genes compared to the bottom sextile when we compare all UTC/SS mutations and just TAA-introducing mutations (P = 0.2107 and P = 0.1172 respectively).

Sextile	Variants	MAPS	N
6 (most constrained)	UTC/SS	0.0483	517
1 (least constrained)	UTC/SS	0.0127	171
6 (most constrained)	TAA	0.1335	144
1 (least constrained)	TAA	0.0311	42

As the power of this analysis is limited by the rarity of UTC / stop-strengthening variants in the current gnomAD release, we also analyzed the proportion of highly conserved bases (phyloP > 2) of potential uORF stop-creating positions compared to matched positions in non-uORF UTR regions. This analysis shows potential UTC introducing positions upstream of more highly constrained genes are also more highly conserved compared to non-uORF positions in constraint- and sequence-matched UTRs. Furthermore, increasing transcript constraint is associated with an increasing proportion of highly conserved UTC-introducing positions. We have included this analysis as **Suppl. Figure 3** in the manuscript.

With regards to UTC gained variants in ClinVar, we would like to highlight a recent study published by Zhang et al. in *Bioinformatics* where > 50 UTC-introducing VUS were annotated in the ClinVar dataset (see Figure 1, Supplemental Table 2)³. The single variant with an assertion of “pathogenic” cited by the reviewer is of a UTC introducing variant leading to gain-of-function that appears to affect a functional peptide controlling downstream translation⁴. Although we state that alternative effects of uORF-disrupting variants on downstream translation are possible in our discussion, we have added language to more explicitly acknowledge this possibility in our discussion and referenced the original study with the following text:

As an example, a pathogenic UTC variant in the *U2HR* gene has previously been reported to confer gain-of-function in Marie Unna hereditary hypotrichosis. However, missense variants in this uORF also confer gain-of-function effects, suggesting that these mutations contribute to pathology through disrupting a functional micropeptide.

2. *I find Figure 1b to be very hard to follow*
 - a. *The ‘All 5’UTR’ point is for some reason in the ‘ncORF’ section*
 - b. *In part ii. The sub-points do not link well to either the legend or the text in the results – I found myself lost as to what most of these points actually represent*
 - c. *If this is not represented by one of the points, the authors should include UTC creating SNVs in non-uORF parts of the UTR.*

We agree with the reviewer that the organization of Figure 1b can be improved. To simplify the figure layout, we have moved the additional matched points to Supplementary Fig. 2. As per the reviewer’s suggestion we have also added a point for UTC creating SNVs in non-uORF parts of the UTR to Figure 1b. We hope this improves the clarity of this figure panel.

3. *The result for CDS-overlap removing vs not variants is not at all convincing given the range of possible values (wide CIs) with the small number of variants assessed – both could have MAPS identical to the all UTC point. Furthermore, calculating significance against the all uORF SNVs point does not make sense to argue this point.*

We appreciate and understand the reviewer's concern and have removed this analysis from the manuscript.

4. *I am unconvinced on multiple aspects of the association analyses as presented. Specifically:*
 - a. *The PMVK variant appears to not be part of the UTR for all GRCh38 transcripts on UCSC – have you confirmed that the effect of this variant is indeed on translation and not through reducing RNA levels? Is this variant an eQTL in GTEx?*

We thank the reviewer for bringing this to our attention. The *PMVK* 5'UTR annotation appears to have changed as of the September 2019 GRCh38 transcript release where the UTR annotation was shortened significantly. We could not find a specific explanation for why this change was made, however it has been previously observed that 5' ends of gene models in reference gene sets are not always complete⁵. Indeed the existence of a longer *PMVK* 5'UTR isoform is supported by transcription start site (TSS) mapping consistent with a longer *PMVK* 5'UTR isoform with a TSS upstream of the current gene model.

Furthermore, we have re-aligned and examined the raw ribosome profiling data from SRP054971⁶ and have confirmed the presence of uniquely-mapped ribosome-protected fragments upstream of the current (shortened) *PMVK* 5'UTR annotation, consistent with evidence that ribosomes are indeed translating this region of the gene. Finally, this exact uORF was also mapped and annotated in a second ribosome profiling dataset from an independent research group in a recent publication⁷.

To clarify this discrepancy in the text, we have added a note to **Table 1** to include the information regarding the updated *PMVK* 5'UTR model, and added supplementary figures (**Suppl. Fig. 7**) showing the upstream transcription start site consistent with the old *PMVK* 5'UTR transcript model, and the raw ribosome profiling reads from SRP054971 mapping 5' to the current 5'UTR *PMVK* annotation.

Finally we have checked the latest GTEx v8 eQTL callset and the *PMVK* SNP is not identified as an eQTL as of the current GTEx release. We have additionally looked to see whether the other phenotype-associated variants are also annotated as cis-eQTLs in GTEx, and note that only *VPS53* is identified in GTEx as a cis-eQTL. We have added these findings to our main text.

- b. *BCL2L13* is only a 5'UTR variant in a minor transcript that doesn't appear to be expressed – on the canonical transcript it is synonymous. This is the only one with any convincing replication data.

In our own analysis of the exon skipping event that differentiates between the primary *BCL2L13* transcript annotation and the alternate minor *BCL2L13* uORF-containing transcript, we find evidence that this transcript is lowly expressed across several cancer contexts (5-10% of RNA-seq reads support exon-skipping), however this proportion can increase significantly in some tumors (see below).

Additionally we note that alternative splicing of the *BCL2* family of genes is a well-documented phenomenon in regulating apoptosis⁸ and that a detailed evaluation of whether lowly-expressed *BCL2L13* isoforms are capable of causing pathology is beyond the scope of the current work. Nevertheless we recognize the reviewer's concern that this variant affects only a "minor" isoform of the *BCL2L13* gene, and have added language in the results to highlight this fact.

- c. *The details of the replication analysis are sparse – the numbers of cases and controls should be displayed along with the exact number of tests i.e. how many diabetes related traits were separately tested?*

We agree with the reviewer that the details surrounding our replication framework can be improved and expanded upon significantly. For replication analyses, we analyzed results based on the exact 4 or 5-digit ICD-9 code identified in the original exploratory analyses and the broader 3 digit ICD-9 code, as previously applied when case numbers in the UKB were insufficient for replication^{7,9}. The criteria for replication was to have the same direction of effect, and $P < 0.05$ for the same 5-digit ICD-9 code. We have updated the method and results section of the manuscript to reflect this approach.

- d. Two of the replication results for PMVK would not survive any multiple testing and I doubt even the UKBB LoF result for this one would survive if this was done properly. The result from all tests of diabetes phenotypes should be shown.

We regret that our framework for the replication analyses was unclear in the original manuscript. For both single-variant and gene-burden replication analyses we sought replication using the 4- or 5-digit ICD-9 code identified in the exploratory analysis in the PMBB. For Phecodes with insufficient cases in the UKB (< 20 cases) the broader 3-digit ICD-9 code was used. As UKB is a healthy population based cohort¹⁰ we did not have sufficient case numbers to directly test for replication for the phenocodes 250.14 and 250.22 so we tested for replication with the broader 3-digit code of 250. Although the broader Phecode for diabetes mellitus was associated with our PMVK variant with a P value of 0.048, re-inspection during this revision showed the direction of effect was discordant. We apologize for this oversight in our original submission. Because of this, we have amended the results and text of our manuscript and are no longer claiming replication for the single variant analysis of PMVK in the UKB. Please find below the full table of cases and controls for each Phecode tested in the replication analysis:

Variant	PheCode	Single Variant				pLOF Gene Burden					
		Cases (UKB)	Controls (UKB)	OR (UKB)	Rep. P (UKB)	OR (UKB)	Rep. P (UKB)	Cases (PMBB)	Controls (PMBB)	OR (PMBB)	Rep. P (PMBB)
rs181302437	250.13	33	32689	3.3E-06	0.986	15.82	0.0073	23	5198	2.46E-05	0.9887
rs181302437	250.14	13	32689	N/A	N/A	N/A	N/A	25	5198	2.48E-05	0.989
rs181302437	250.22	11	32689	N/A	N/A	N/A	N/A	315	6134	5.96E-05	0.9672
rs181302437	250	1905	32689	0.44	0.0483	0.81	0.7219	2772	6143	0.82	0.7621
rs35915949	300.1	627	31247	0.88	0.1631	1.35	0.6751	1060	6939	1.32	0.8052
rs35915949	300	684	31247	0.88	0.1757	1.48	0.5833	1249	6939	0.66	0.5808
rs139848407	270.33	11	34565	N/A	N/A	N/A	N/A	30	7727	7.25E-06	0.9892
rs139848407	270	34	34554	9.35E-06	0.9887	9.74	0.0264	134	9594	4.42	0.1518
rs140799351	610	277	33848	1.77E-04	0.973	N/A	N/A	55	7689	N/A	N/A
rs140799351	187	51	33957	56.02	0.0003	N/A	N/A	26	7700	N/A	N/A
rs140799351	187.2	30	33957	79.78	0.0002	N/A	N/A	34	7700	N/A	N/A
rs28365863	527	N/A	N/A	N/A	N/A	N/A	N/A	90	9774	2.11	0.0055
rs116450723	350	N/A	N/A	N/A	N/A	N/A	N/A	362	9415	2.35E-05	0.9659

We have included this as a supplementary table (**Suppl. Table 4**) in the manuscript. We also state clearly in the text that the abnormal fasting blood glucose result is nominally significant and we are not claiming that this result is a replication of the original PMVK finding. To make this point more clear we have removed it from Table 1.

Finally with regards to the reviewer's comments on adjusting for multiple testing, we note that we are performing hypothesis-driven tests of association between the genetic variant and a phenotype captured by highly correlated ICD-9 / 10 codes in electronic health records which are primarily used for billing purposes. For this reason, in the replication analyses we test for association with the original phenotypes and related Phecodes. Since the phenotype is captured by several related ICD codes, the most appropriate method for correcting P-values is not clear. In a review of previous PheWAS literature we note that unadjusted P-values for replication are commonly reported⁹⁻¹². To address the reviewer's concern, we have included the following text in our discussion regarding the challenge of P-value adjustment in PheWAS in the context of replication studies:

Finally, we note that being a hospital-based biobank, participants in the PMBB are generally less healthy than the general population. As phenotypes within broader disease Phecode families are often highly correlated, we sought to replicate associations uncovered in the discovery analysis by first testing for a specific hypothesis-driven phenotype association in addition to related phenotypes in the corresponding Phecode families. We recognize that controlling for Type 1 error in this framework remains challenging, however to remedy this we sought additional confidence by further replicating significant uORF-variant associations through loss-of-function gene-burden analyses.

- e. *The LoF replication analysis is misleading – on reading the methods it seems that missense variants with high REVEL scores were also included. This is not the LoF test that is claimed in the main text and this should be made clear. How many of these associations are driven by missense variants rather than LoF? Was there any filter on variant frequency (i.e. as common pLoFs are likely not true LoF)?*

We agree with the reviewer that the details of our approach as described in the main text can be improved. We have clarified that the LoF replication analysis includes missense variants with high REVEL scores. We used a variant frequency filter $MAF \leq 0.1\%$ and have updated the methods section to include this information.

As our hypothesis was that any particularly rare, high REVEL-scored missense variant or pLOF variant could lead to loss-of-function in this gene-burden analysis, it is unclear to us the utility of identifying which associations are specifically driven by missense variants compared to LOF. Since each of these variants are rare, it is difficult to draw conclusions about whether single variants are driving the association for each gene-burden test.

- f. The authors state “A second uORF-disease association was replicated for NALCN and the parent PheCode of disorders of plasma protein metabolism in the UKB” – with a p-value of only 0.026 and no multiple-testing correction this claim is untrue.

We regret that our original text was unclear on our replication framework. The original association of NALCN was with amyloidosis. As there were only 11 cases in the UKB cohort we are underpowered to test for association between gene burden and the specific amyloidosis PheCode. Consequently we tested for replication with the parent PheCode of “disorders of plasma protein metabolism”. With regards to multiple-testing correction we refer the reviewer to our response above (4d).

5. The functional assays would be more convincing if the authors showed the effect of simultaneously removing the uORF start and the stop strengthening variant. Also, why did the authors not assess the effect of the BCL2L13 variant which is the only one with convincing replication data? I imagine this may be because this work was done upstream of, or in tandem with the replication studies.

We have included luciferase assay results for the BCL2L13 variant and added it to Fig 5:

With regards to simultaneously removing the uORF start codon and the stop strengthening variant, we have reservations about the added utility of testing

luciferase constructs with both mutations. It is possible that these uORFs are capable of initiating translation using alternative uORF start codons (but sharing the same stop codon) that were not originally identified using the RibORF algorithm. If this is the case then we could still observe a relative decrease in CDS expression for the simultaneous uORF-KO and stop-strengthened construct compared to the uORF-KO construct alone. Because of these interpretation challenges, we elected to pursue the focused set of mutant constructs described herein.

6. *In the discussion the authors claim “These findings establish that uORF UTC and stop-strengthening variants can have functional consequences on gene expression and cause disease in humans” – this is not supported by the data shown. The authors show that these variants may be associated with phenotypes but not that they can cause disease.*

We agree with the reviewer that our results support that these variants are associated with phenotypes but not that they can cause disease. We have amended the text to make more clear that these variants are associated with phenotypes.

7. *The analysis in the discussion and presented in the supplementary note includes a lot of assumptions and is hard to assess as included – this should either be included and displayed properly as an analysis in the paper or removed.*

We agree with the reviewer that the analysis in the discussion and supplementary notes are difficult to assess as included, however because this analysis is not directly related to the primary conclusions of the manuscript we believe that keeping it in the discussion is appropriate. To improve the interpretability of this analysis, we have reframed the text to more clearly emphasize the assumptions we are making in addition to the analysis results.

Minor:

1. *The first sentence of the abstract downplays what is already known about uORFs – there has been plenty of prior research showing the negative role of these elements on translation, but this sentence suggests their function is unknown.*

We agree with the reviewer that the language used in this sentence is confusing. We have amended the first sentence of the abstract to refer to non-canonical upstream open reading frames more generally rather than uORFs specifically.

2. *The final sentence of the abstract states “translated uORFs are genetically constrained regulatory elements in 40% of human genes” – where does 40% come from? The*

remainder of the paper uses 50%. Also, this sentence is misleading as it suggests the sequence of the uORFs are constrained, which they show is not the case.

We clarified this sentence in the abstract.

- 3. The introduction is rather short and a lot of what could be in the introduction is within the results, inflating the size of these sections. For example, there is nothing in the intro about what is currently known about the lack of conservation across uORF sequences. Also, most of the first paragraph of the results would be better placed in the introduction. The introduction also only states that prior work by Whiffin et al. looked at creating new uORFs when they also assessed uORF stop removing variants.*

We have re-organized the first section of the results and introduction to follow the suggested structure by the reviewer.

- 4. How many genes are represented by translated uORFs? How many genes have multiple?*

We note throughout the manuscript that we are using uORFs mapped from 4392 genes (from 10,946 expressed genes) identified by deep ribosome profiling of two human cell lines from the Ji et al. 2015 eLife publication. These data are also displayed in Suppl. Figure 1. Of these 4392 genes 1466 have multiple uORFs (> 1) mapped to their 5'UTRs.

- 5. How does the enrichment in Figure 1c compare to stop codons used in canonical coding sequences?*

While we agree with the reviewer that this would be an interesting additional analysis question we feel that evaluating stop codon usage across protein-coding sequences is beyond the scope of the current manuscript. For a previous analysis regarding this topic we point the reviewer to the relevant work by Trotta and colleagues¹³.

- 6. It would be helpful to see an 'opposite' point to TAA gained UTC – is there any selection against TGA gained UTCs?*

We have performed an analysis of MAPS scores for TGA-gained UTCs. The MAPS scores for these variants is 0.0216 (n = 1371, 95% CI: 0.0129 - 0.0471) which is not significantly higher than MAPS scores for all uORF variants (MAPS = 0.0138, P = 0.2833). We have included this in the manuscript results text.

- 7. Conservation was used in Whiffin et al. to assess stop-removing variants not start creating variants – the text on this should be corrected*

The Whiffin et al. paper examined both start creating variants and stop-removing variants. We have included a sentence in the introduction acknowledging this previous work as follows:

Previous analyses of large-scale population data have shown that genetic variants creating new uORFs are rare, suggesting that these variants are subjected to strong negative selection due to their capacity to cause pathogenic loss-of-function of associated proteins^{14,15}. Moreover, it has been shown that variants destroying stop codons in translated uORFs are under strong negative selection, presumably because the resultant translational readthrough can decrease translation initiation at the coding sequence (CDS)¹⁶.

8. *Figure 2c would benefit from an ‘all uORF’ point*

We have added an “all uORF” point to Figure 2c for comparison.

9. *It would be informative to reference Schulz et al. Sci Reports 2018 when discussing start removing variants*

We have included a reference to Schulz et al. Sci Reports in our discussion of start removing variants.

10. *Are stop and start matched variants also matched by UTR position?*

Stop and start matched variants are matched by UTR position relative to the downstream annotated CDS for the phyloP conservation analysis but not for the MAPS analysis which uses uORF variants matched by underlying trinucleotide context that do not disrupt uORF start codons / create new UTCs. We have clarified this in the methods.

11. *“Using exome sequencing from the PMBB, we identified heterozygous and homozygous individuals carrying mutations in uORFs which introduce upstream termination codons and stop-strengthening mutations. For the latter class, we focused on variants that introduced TAA stop codons, as the heightened MAPS score for such variants implied these mutations would be most deleterious” – I assume this should say “For the former class”?*

We thank the reviewer for pointing out this error and have corrected it in the manuscript. The text now reads “For the former class”.

12. *Table 1 should report variant frequencies not just numbers for cases and controls. There should also be full details on variant numbers for all replication analyses. This should include split data for Europeans and Africans for PMBB.*

We have included a supplemental table (**Suppl. Table 3**) with minor allele frequencies for the association studies presented.

Gene	Carrier Freq (PMBB)	PMBB EUR	PMBB AFR	Carrier Freq (UKB)	LOF Carrier Freq (PMBB)	LOF Carrier Freq (UKB)
PMVK	0.00298	0.00336	0.00000	0.00324	0.00086	0.00094
VPS53	0.09468	0.10954	0.01940	0.14370	0.00131	0.00109
NALCN	0.00147	0.00129	0.00106	0.00084	0.00122	0.00162
BCL2L13	0.00028	0.00039	0.00190	0.00029	0.00028	0.00025
SHMT2	0.01266	0.00540	0.03715	0.00749	0.00182	0.00033
MOAP1	0.00188	0.00006	0.00793	N/A	0.00026	N/A

13. *More details should be given on the definition of 5'UTRs*

We have included text in the methods section specifying that 5' and 3' UTR definitions are derived from the Ensembl v86 transcript annotations.

14. *The MAPS results for both missense and LoF variants look to be depleted compared to previous work – this may be because the authors deviated from just using the effect on the canonical transcript to, for missense, removing variants with a coding annotation on any transcript, and, for pLoF, including start and stop lost which are not always true LoF.*

We agree that this could be due to differences between variant sets used to calculate MAPS scores. We stated in the methods that the set of variants used to calculate MAPS scores for pLoF variants included those with an annotation of stop_lost and start_lost. We also note that we are using the newer v3 release of gnomAD which contains more individuals than previous publications that report MAPS scores using the gnomAD v2 release or exomes.

Reviewer #2 (Remarks to the Author):

This clearly-structured and well-written manuscript from David Lee and colleagues builds upon recent work (most notably from Nicky Whiffin) exploring the degree of purifying selection ("constraint") on 5' UTRs. The rationale for studying these regions is manifold: first translation regulation is a deeply complex and relatively poorly understood process; second, less than half of individuals with rare disease receive a genetic diagnosis and the variants in 5'UTRs are largely ignored owing to a focus on exome sequencing and difficulty prioritizing such variants. Overall, I found this to be a very compelling and string study that focuses on specific classes of

5'UTR variation to which special attention should now be paid in disease studies. I have two concerns that I would like the authors to address.

1. As you know, many genes have multiple isoforms whose 5'UTRs start at different genomic positions. As a result, sequence that is part of a 5'UTR in one isoforms could be CDS in another. It was not clear to me how you accounted for the possibility that genetic constraint predicted through higher MAPS scores manifests from purifying selection on CDS in another isoform?

We thank the reviewer for bringing this potential confounding factor in our analysis to our attention. The majority of uORFs mapped in the original Ji et al. publication (from which we derived our ncORF annotations) were identified outside of protein coding sequences⁶. In order for the higher MAPS scores in uORF variants to reflect purifying selection on the CDS in another isoform they must, on average, be more deleterious than synonymous variants in these protein-coding regions (since the MAPS model is calibrated using synonymous coding variants as a baseline for MAPS = 0). With this in mind, we have re-analyzed our uORF UTC / stop-strengthening variant sets to identify uORF positions that may potentially overlap with the annotated CDS of any other mRNA. A summary of this analysis is presented in the table below:

uORF Variant Type	Number of SNVs	SNVs overlapping any CDS (%)	Original MAPS (95% CI)	MAPS with CDS-overlap removed (95% CI)
UTC	3024	163 (5.39%)	0.0373 (0.0194-0.0552)	0.0362 (0.0172-0.0552)
TAA gained	928	93 (10.02%)	0.0726 (0.0407-0.1045)	0.0765 (0.0430-0.1101)
Stop > TAA	499	51 (10.22%)	0.0971 (0.0537-0.1404)	0.1051 (0.0595-0.1507)

This analysis shows that potential CDS-overlapping uORF variants make up a small proportion of each set of variants tested, and their inclusion has minimal influence over MAPS scores. We have included this analysis as **Suppl. Table 1** and acknowledged the minimal impact of removing CDS-overlapping uORF variants in the main text.

2. Previous work this year from your group showed constraint on G-quadruplexes in 5' UTRs. I was surprised to not see an evaluation of the constraint you observe in this analysis with respect to the presence or absence of a G-quadruplex, as that paper similarly argues that "negative selection acting on central guanines of UTR pG4s is comparable to that of missense variation in protein-coding sequences." To what degree is the constraint you observe on stop-introducing and stop-strengthening mutations in translated uORFs a result, at least in part, on constraint owing to the presence of a G-quadruplex? I am curious because the prior paper makes the argument that such

regions are enriched for cis-eQTLs and RBP interactions, which is a completely distinct mechanistic explanation for constraint on these sequences.

Although our previous work focused on constraints involving putative G-quadruplex forming sequences in UTRs, we found by MAPS analysis that enrichment for rare variants only occurred for the central G-quadruplex guanine in each trinucleotide G-run. For our analysis of uORF stop-introducing / stop-strengthening mutations to be confounded by the presence of a G-quadruplex, the stop-introducing mutant would also have to disrupt the central guanine position of a canonical G-quadruplex-forming sequence. The only mutation affecting the central guanine of a predicted G-quadruplex forming sequence (GGG) that could also create stop-introducing mutations is a G>A mutation (for **TGGG** > **TGAG**). Repeating UTC analysis with all of these possible mutations removed (n = 57), we continue to observe significant enrichment over baseline (MAPS = 0.0377, 95% CI: 0.0196-0.0557). Thus the potential influence of G-quadruplex forming sequences on our calculation for MAPS scores for uORF-UTC variants is minimal.

We have added text in the Results to address the reviewer's concerns and summarize results of our MAPS analyses with possible CDS-overlapping or G-quadruplex disrupting variants removed:

To account for the possibility that the heightened MAPS scores for UTC introducing variants resulted from overlap between 5'UTRs and annotated coding sequences in different mRNA isoforms, we repeated this analysis excluding all uORF variants overlapping with an annotated CDS sequence. Re-calculated MAPS scores with all CDS-overlapping variants removed remained essentially unchanged, ruling out the possibility that the enrichment in rare variation for UTC-creating SNVs is driven by negative selection on coding sequences (**Suppl. Table 1**). Additionally, we have previously observed that variants destroying the central guanine of putative G-quadruplex forming sequences exhibit heightened MAPS scores in UTRs. We repeated this analysis with all potential G-quadruplex disrupting variants (n = 57) excluded, seeing a negligible effect on MAPS scores for all UTC-creating variants (MAPS = 0.0377, 95% CI: 0.0196-0.0557). Overall, the strong selective pressure to remove uORF UTC-creating SNVs implies that these variants are also more likely to have functional biological consequences.

Minor

1. *The x-axis for Fig3A is unlabelled and thus confusing.*

We thank the reviewer for pointing out this issue and have added a label to the x-axis of Fig3A.

Reviewer #3 (Remarks to the Author):

In this work, the authors nicely describe classes of variants within the 5'UTR that punitively influence translational efficiency by disrupting upstream open reading frames. The authors examine both start and termination codons (and their relative strengths) within these uORFs and examine them from a conservation and evolutionary perspective. They find selective pressure preserving the general structure of the uORF, however the amino acid content was not found to be under the same degree of constraint as missense variants within coding sequence. The authors then select rare variants creating premature stop codons within the uORF of a few genes within large biobank studies to conduct a phenome-wide association study to identify their potential phenotypic consequences. In general this study is well-conceived, constructed, and executed.

My primary concern with the manuscript is the rather confusing way the authors have used "gene expression" and "protein expression" interchangeably in both wording and some aspects of the experimental design. For example, in the first paragraph of the introduction, the authors state "Specific uORFs are known to control gene expression by tuning translation rates of downstream protein-coding sequences". I believe here the authors mean "protein expression". This happens throughout the manuscript, though I believe the primary focus of this work is on protein expression and the translation process. This is further exemplified by the framing of the luciferase reporter assay as determining "if these variants could affect gene expression", when in fact the authors are using this system to directly measure protein product translated from different UTR-based constructs. Clarifying this language will make the focus of the manuscript much more clear and will prevent lots of confusion.

We thank the reviewer for this suggestion and have clarified the language throughout to emphasize protein expression over gene expression.

I will also admit to some disappointment that ribosomal profiling experiments were not attempted, especially given the availability of the luciferase construct. This would have provided additional mechanistic evidence of the role of the uORFs in ribosomal positioning, and would have allowed some investigation of the changes in stop-codon strength and the other factors the authors mention in the limitations section.

The focus of the current manuscript is on investigating patterns of selection in uORFs and exploring possible functional consequences of uORF-disruptions through mutations introducing or strengthening stop codons. That said, we are very interested in investigating mechanisms of ribosome loading on coding sequences as a function of uORF stop codon context, length, and other factors. We hope that our findings provide fertile ground for future studies.

A minor concern - the information in table 1 is difficult to read given the column widths - I would convert this to a landscape page.

We have reformatted the table to improve readability.

References:

1. McCabe, M. J. *et al.* Novel application of luciferase assay for the in vitro functional assessment of KAL1 variants in three females with septo-optic dysplasia (SOD). *Mol. Cell. Endocrinol.* **417**, 63–72 (2015).
2. Babu, D. *et al.* Variants in the 5'UTR reduce SHOX expression and contribute to SHOX haploinsufficiency. *European Journal of Human Genetics* (2020)
doi:10.1038/s41431-020-0676-y.
3. Zhang, X., Wakeling, M., Ware, J. & Whiffin, N. Annotating high-impact 5'untranslated region variants with the UTRannotator. doi:10.1101/2020.06.03.132266.
4. Wen, Y. *et al.* Loss-of-function mutations of an inhibitory upstream ORF in the human hairless transcript cause Marie Unna hereditary hypotrichosis. *Nat. Genet.* **41**, 228–233 (2009).
5. Abugessaisa, I. *et al.* refTSS: A Reference Data Set for Human and Mouse Transcription Start Sites. *J. Mol. Biol.* **431**, 2407–2422 (2019).
6. Ji, Z., Song, R., Regev, A. & Struhl, K. Many lncRNAs, 5'UTRs, and pseudogenes are translated and some are likely to express functional proteins. *Elife* **4**, e08890 (2015).
7. Martinez, T. F. *et al.* Accurate annotation of human protein-coding small open reading frames. *Nat. Chem. Biol.* **16**, 458–468 (2020).
8. Akgul, C., Moulding, D. A. & Edwards, S. W. Alternative splicing of Bcl-2-related genes: functional consequences and potential therapeutic applications. *Cell. Mol. Life Sci.* **61**, 2189–2199 (2004).
9. Verma, A. *et al.* eMERGE Phenome-Wide Association Study (PheWAS) identifies clinical associations and pleiotropy for stop-gain variants. *BMC Med. Genomics* **9 Suppl 1**, 32 (2016).

10. Fry, A. *et al.* Comparison of Sociodemographic and Health-Related Characteristics of UK Biobank Participants With Those of the General Population. *Am. J. Epidemiol.* **186**, 1026–1034 (2017).
11. Locke, A. E. *et al.* Exome sequencing of Finnish isolates enhances rare-variant association power. *Nature* **572**, 323–328 (2019).
12. Cai, T. *et al.* Association of Interleukin 6 Receptor Variant With Cardiovascular Disease Effects of Interleukin 6 Receptor Blocking Therapy: A Phenome-Wide Association Study. *JAMA Cardiol* **3**, 849–857 (2018).
13. Trotta, E. Selective forces and mutational biases drive stop codon usage in the human genome: a comparison with sense codon usage. *BMC Genomics* **17**, 366 (2016).
14. Whiffin, N. *et al.* Characterising the loss-of-function impact of 5' untranslated region variants in 15,708 individuals. *bioRxiv* 543504 (2019) doi:10.1101/543504.
15. Calvo, S. E., Pagliarini, D. J. & Mootha, V. K. Upstream open reading frames cause widespread reduction of protein expression and are polymorphic among humans. *Proc. Natl. Acad. Sci. U. S. A.* **106**, 7507–7512 (2009).
16. Whiffin, N. *et al.* Characterising the loss-of-function impact of 5' untranslated region variants in 15,708 individuals. *Nature Communications* vol. 11 (2020).

Reviewer #1 (Remarks to the Author):

This study by Lee et al. assesses the impact of variants that disrupt start and stop codons of translated upstream open reading frames using signals of negative selection from >70,000 individuals in gnomAD. The authors identify variants creating stop variants in uORF sequences and that strengthen uORF stop codons that look to be deleterious. Finally, the authors use biobank datasets to look for association of these variants with phenotypes.

My opinions on this paper are a tale of two halves. I think the first half, which uses MAPS scores to assess the deleterious of stop gain and strengthening variants is elegant and interesting. However, the second half of the paper on phenotype associations is weak and unconvincing. I also think the authors make some strong claims (such as a 'loss-of-function' effect) that are not supported by their data. Below are my specific comments.

Major:

1. The assertion that these variants can cause LoF is not supported by the data. Yes, they may often (but perhaps not always) reduce protein translation, however there is no evidence that this will cause a complete LoF (and therefore be useful for clinical variant interpretation). This should therefore not be stated in the title, abstract or conclusions. Specifically:
 - a. Whilst MAPS scores are increased, this does not suggest a mechanism.
 - b. The reporter assays show reduced protein levels, but not with a large effect – none of the tested examples reduce protein levels by more than ~40%.
 - c. The authors do not report example disease-causing variants (as opposed to associations)
 - d. The authors could add support to this hypothesis by looking if MAPS scores are enriched upstream of LoF intolerant genes (low LOEUF scores)
 - e. Looking at UTC gained variants in ClinVar, there seems to be only a single example, and this seems to be associated with up-regulation of protein expression.
2. I find Figure 1b to be very hard to follow
 - a. The 'All 5'UTR' point is for some reason in the 'ncORF' section
 - b. In part ii. The sub-points do not link well to either the legend or the text in the results – I found myself lost as to what most of these points actually represent
 - c. If this is not represented by one of the points, the authors should include UTC creating SNVs in non-uORF parts of the UTR.
3. The result for CDS-overlap removing vs not variants is not at all convincing given the range of possible values (wide CIs) with the small number of variants assessed – both could have MAPS identical to the all UTC point. Furthermore, calculating significance against the all uORF SNVs point does not make sense to argue this point.
4. I am unconvinced on multiple aspects of the association analyses as presented. Specifically:
 - a. The PMVK variant appears to not be part of the UTR for all GRCh38 transcripts on UCSC – have you confirmed that the effect of this variant is indeed on translation and not through reducing RNA levels? Is this variant an eQTL in GTEx?
 - b. BCL2L13 is only a 5'UTR variant in a minor transcript that doesn't appear to be expressed – on the canonical transcript it is synonymous. This is the only one with any convincing replication data.
 - c. The details of the replication analysis are sparse – the numbers of cases and controls should be displayed along with the exact number of tests i.e. how many diabetes related traits were separately tested?
 - d. Two of the replication results for PMVK would not survive any multiple testing and I doubt even the UKBB LoF result for this one would survive if this was done properly. The result from all tests of diabetes phenotypes should be shown.
 - e. The LoF replication analysis is misleading – on reading the methods it seems that missense variants with high REVEL scores were also included. This is not the LoF test that is claimed in the main text and this should be made clear. How many of these associations are driven by missense variants rather than LoF? Was there any filter on variant frequency (i.e. as common pLoFs are likely not true LoF)?
 - f. The authors state "A second uORF-disease association was replicated for NALCN and the parent

PheCode of disorders of plasma protein metabolism in the UKB" – with a p-value of only 0.026 and no multiple-testing correction this claim is untrue.

5. The functional assays would be more convincing if the authors showed the effect of simultaneously removing the uORF start and the stop strengthening variant. Also, why did the authors not assess the effect of the BCL2L13 variant which is the only one with convincing replication data? I imagine this may be because this work was done upstream of, or in tandem with the replication studies.

6. In the discussion the authors claim "These findings establish that uORF UTC and stop-strengthening variants can have functional consequences on gene expression and cause disease in humans" – this is not supported by the data shown. The authors show that these variants may be associated with phenotypes but not that they can cause disease.

7. The analysis in the discussion and presented in the supplementary note includes a lot of assumptions and is hard to assess as included – this should either be included and displayed properly as an analysis in the paper or removed.

Minor:

1. The first sentence of the abstract downplays what is already known about uORFs – there has been plenty of prior research showing the negative role of these elements on translation, but this sentence suggests their function is unknown.

2. The final sentence of the abstract states "translated uORFs are genetically constrained regulatory elements in 40% of human genes" – where does 40% come from? The remainder of the paper uses 50%. Also, this sentence is misleading as it suggests the sequence of the uORFs are constrained, which they show is not the case.

3. The introduction is rather short and a lot of what could be in the introduction is within the results, inflating the size of these sections. For example, there is nothing in the intro about what is currently known about the lack of conservation across uORF sequences. Also, most of the first paragraph of the results would be better placed in the introduction. The introduction also only states that prior work by Whiffin et al. looked at creating new uORFs when they also assessed uORF stop removing variants.

4. How many genes are represented by translated uORFs? How many genes have multiple?

5. How does the enrichment in Figure 1c compare to stop codons used in canonical coding sequences?

6. It would be helpful to see an 'opposite' point to TAA gained UTC – is there any selection against TGA gained UTCs?

7. Conservation was used in Whiffin et al. to assess stop-removing variants not start creating variants – the text on this should be corrected

8. Figure 2c would benefit from an 'all uORF' point

9. It would be informative to reference Schulz et al. Sci Reports 2018 when discussing start removing variants

10. Are stop and start matched variants also matched by UTR position?

11. "Using exome sequencing from the PMBB, we identified heterozygous and homozygous individuals carrying mutations in uORFs which introduce upstream termination codons and stop-strengthening mutations. For the latter class, we focused on variants that introduced TAA stop codons, as the heightened MAPS score for such variants implied these mutations would be most deleterious" – I assume this should say "For the former class"?

12. Table 1 should report variant frequencies not just numbers for cases and controls. There should also be full details on variant numbers for all replication analyses. This should include split data for Europeans and Africans for PMBB.

13. More details should be given on the definition of 5'UTRs

14. The MAPS results for both missense and LoF variants look to be depleted compared to previous work – this may be because the authors deviated from just using the effect on the canonical transcript to, for missense, removing variants with a coding annotation on any transcript, and, for pLoF, including start and stop lost which are not always true LoF.

Reviewer #2 (Remarks to the Author):

This clearly-structured and well-written manuscript from David Lee and colleagues builds upon recent work (most notably from Nicky Whiffin) exploring the degree of purifying selection ("constraint") on 5' UTRs. The rationale for studying these regions is manifold: first translation regulation is a deeply complex and relatively poorly understood process; second, less than half of individuals with rare disease receive a genetic diagnosis and the variants in 5'UTRs are largely ignored owing to a focus on exome sequencing and difficulty prioritizing such variants. Overall, I found this to be a very compelling and string study that focuses on specific classes of 5'UTR variation to which special attention should now be paid in disease studies. I have two concerns that I would like the authors to address.

1. As you know, many genes have multiple isoforms whose 5'UTRs start at different genomic positions. As a result, sequence that is part of a 5'UTR in one isoform could be CDS in another. It was not clear to me how you accounted for the possibility that genetic constraint predicted through higher MAPS scores manifests from purifying selection on CDS in another isoform?

2. Previous work this year from your group showed constraint on G-quadruplexes in 5' UTRs. I was surprised to not see an evaluation of the constraint you observe in this analysis with respect to the presence or absence of a G-quadruplex, as that paper similarly argues that "negative selection acting on central guanines of UTR pG4s is comparable to that of missense variation in protein-coding sequences." To what degree is the constraint you observe on stop-introducing and stop-strengthening mutations in translated uORFs a result, at least in part, on constraint owing to the presence of a G-quadruplex? I am curious because the prior paper makes the argument that such regions are enriched for cis-eQTLs and RBP interactions, which is a completely distinct mechanistic explanation for constraint on these sequences.

Minor

1. The x-axis for Fig3A is unlabelled and thus confusing.

Reviewer #3 (Remarks to the Author):

In this work, the authors nicely describe classes of variants within the 5'UTR that punitively influence translational efficiency by disrupting upstream open reading frames. The authors examine both start and termination codons (and their relative strengths) within these uORFs and examine them from a conservation and evolutionary perspective. They find selective pressure preserving the general structure of the uORF, however the amino acid content was not found to be under the same degree of constraint as missense variants within coding sequence. The authors then select rare variants creating premature stop codons within the uORF of a few genes within large biobank studies to conduct a phenome-wide association study to identify their potential phenotypic consequences. In general this study is well-conceived, constructed, and executed.

My primary concern with the manuscript is the rather confusing way the authors have used "gene expression" and "protein expression" interchangeably in both wording and some aspects of the experimental design. For example, in the first paragraph of the introduction, the authors state "Specific uORFs are known to control gene expression by tuning translation rates of downstream protein-coding sequences". I believe here the authors mean "protein expression". This happens throughout the manuscript, though I believe the primary focus of this work is on protein expression and the translation process. This is further exemplified by the framing of the luciferase reporter assay as determining "if these variants could affect gene expression", when in fact the authors are using this system to directly measure protein product translated from different UTR-based constructs. Clarifying this language will make the focus of the manuscript much more clear and will prevent lots of

confusion.

I will also admit to some disappointment that ribosomal profiling experiments were not attempted, especially given the availability of the luciferase construct. This would have provided additional mechanistic evidence of the role of the uORFs in ribosomal positioning, and would have allowed some investigation of the changes in stop-codon strength and the other factors the authors mention in the limitations section.

A minor concern - the information in table 1 is difficult to read given the column widths - I would convert this to a landscape page.

REVIEWER COMMENTS

Reviewer #1 (Remarks to the Author):

I would first like to thank the authors for their very careful and detailed responses to my comments.

A remaining major concern is the authors use of loss-of-function. The authors claim to have diminished these claims, but I can see very little evidence of this in the manuscript, apart from some minor re-wording of the title. The authors present evidence that UTC-creating and stop strengthening variants can reduce protein expression, but whether this equates to full loss-of-function is not clear. I believe this term should therefore be removed from the title. The term "loss-of-function" can also be easily switched with "reduced translation" in the abstract. Finally, the first sentence of the discussion should similarly be edited to "capable of reducing downstream protein expression." or similar. This is a very important distinction for anyone interpreting these variants for roles in disease.

We thank the reviewer for this comment. To address the reviewer's concerns we have exchanged loss-of-function when referring to uORF stop-gain or strengthening variants for "reduced translation". Additionally we have removed "loss-of-function" from the title of the manuscript.

A second major concern regards the analysis in the discussion and supplementary note - I do not know why this is included, and the numbers are, I believe, misleading. "If we assume that pathogenic UTC or stop- strengthening mutations are under similar selective pressures as pathogenic loss-of-function variants in protein-coding regions of the genome" is a very strong assumption that is not supported by the data. The effect of these variants in terms of the level of reduction of protein levels is going to be very gene dependent. Similarly, not all genes are known to cause disease through a LoF mechanism. Similarly, I don't understand the assumption for missense variants given the authors show these are not constrained. It is my view that these analyses should be removed from the manuscript.

The reviewer's comment is appreciated. We have removed this analysis from the manuscript.

I also still have the following remaining more minor comments:

- Do the authors have a hypothesis for the increased MAPS of 5'UTR stop-gained not in uORFs (over all 5'UTR variants) - is this signal due to uORF sequences not found in the riboseq data?

We agree with the reviewer that it is interesting that 5'UTR stop-gain variants appear to be enriched for singletons over all 5'UTR variants. As we are only using 5'UTR uORFs mapped from two cell lines in one ribosome profiling experiment for our analyses, we believe that this heightened MAPS score could reflect the presence of uORFs in other 5'UTR sequences that were not experimentally mapped in the original Ji. et al. data from which our ORF annotations are derived. Indeed uORF usage has previously been shown to differ between cell types and be dynamically regulated in response to extracellular

stimuli so it is likely that only a subset of translation-capable uORFs were captured across these two cell lines.

- The authors have changed variant to mutation throughout the manuscript. This is not in line with best practices as mutation has connotations of pathogenicity. All occurrences should be switched back to variant.

We thank the reviewer for bringing this to our attention and have changed “mutation” back to variant throughout the manuscript.

- The ncORF stop-gained label in figure 1b is hard to interpret for anyone not so familiar with these data
- this should be better linked to the main text and legend text.

We agree with the reviewer that the label can be better explained. We have updated the Figure legend and the main text in the results to clarify our use of the “ncORF” label.

- I am not sure where sup table 4 does not have case and control numbers for the UKB LoF.

Case and control numbers for UKB LoF analyses are the same for case and control numbers for the single variant analysis. We have reorganized **Suppl. Table 4** to display the data more clearly.

- Given the authors claim of new gene-disease relationships in the discussion, it would be helpful to include in Table 1 whether or not the gene has previously been associated with a related phenotype. For e.g. from my limited search, it appears that BCL1L13 is known to be linked to cancer.

We have performed a literature search for each of the gene-disease associations reported in our study and to our knowledge only BCL2L13 has previously been implicated in cancer. To maintain the focus of Table 1 on reporting our disease-gene association statistics, we have not included this information in Table 1. Instead we have expanded our discussion to explore whether the identified genes have been previously associated with related phenotypes. Interestingly our research uncovered a recent publication reporting the involvement of SHMT2 in a clinical syndrome characterized by cardiac and movement disorders which may possibly be related to our findings of salivary gland disorders. Upon further review of additional phenotypes that were nominally associated with the SHMT2 uORF stop strengthening variant identified in this study we uncovered a constellation of cardiac and movement-related Phecodes at $P < 0.05$. We have added text to our Discussion describing these possibly related findings:

Of the new variant-phenotype associations uncovered by our study, only the association between *BCL2L13* and cancer has been previously reported^{1,2}. Interestingly, bi-allelic loss-of-function mutations in *SHMT2* have recently been described in a novel brain and heart developmental syndrome involving spastic paraparesis and ataxias⁵⁴. Indeed, in

addition to the phenome-wide significant association with diseases of the salivary gland uncovered in our study, the SHMT2 uORF stop-strengthening variant was nominally associated with several Phecodes related to cardiac and movement disorders in the PMBB (**Suppl. Table 6**), including Congenital anomalies of the great vessels (ICD 747.13, P = 0.0117), Abnormal involuntary movements (350.1, P = 0.0238), Abnormality of gait (350.2, P = 0.02575), Mobitz II AV block (426.22, P = 0.03432), and Arrhythmia (cardiac) NOS (427.5, P = 0.04977). These additional nominal associations suggest that SHMT2 uORF variants may be capable of contributing to similar phenotypic consequences as described in loss-of-function mutation carriers, however further studies are needed to investigate this possibility.

Additionally, we have included **Suppl. Table 6** to summarize these findings:

Suppl. Table 6: Nominal cardiac and movement disorder associations with SHMT2 stop-strengthening variant uncovered through PheWAS in Penn Medicine Biobank.

phenotype	beta	OR	SE	p	n_total	n_cases	n_controls	allele_freq	description
747.13	1.518	4.561	0.602	0.0117	7300	62	7238	0.005410959	Congenital anomalies of great vessels
350.1	1.124	3.077	0.497	0.0238	9519	104	9415	0.011503309	Abnormal involuntary movements
350.2	0.660	1.935	0.296	0.0258	9626	211	9415	0.011790983	Abnormality of gait
426.22	1.635	5.129	0.773	0.0343	2481	50	2431	0.005441354	Mobitz II AV block
972.1	1.434	4.197	0.703	0.0412	6285	40	6245	0.005966587	Cardiac rhythm regulators causing adverse effects in therapeutic use
427.5	0.859	2.362	0.438	0.0498	3412	157	3255	0.013335287	Arrhythmia (cardiac) NOS

- In my opinion, the authors still do not really discuss the potential for UTC variants to also be gain of function and how this is likely gene dependent. They introduce a single example as an "additional factor" when it could be an alternative mechanism.

We have increased our emphasis on possible UTC-variant gain-of-function effects by adding the following text in our discussion:

Indeed, previous studies have shown that a multitude of factors may impact uORF regulatory function, and it is likely that UTC variants are also capable of causing gain-of-function in some uORF-regulated genes.

- Given the conflicting direction of the PMVK association in the UKB analysis this should be made clear in Table 1 - it is still listed as "Yes" for replication, which is misleading. Some indication of the direction of effect should be included in the table.

We have amended Table 1 so that the PMVK single-variant analysis in the UKB now reads "No".

- The authors state that "We have clarified that the LoF replication analysis includes missense variants with high REVEL scores". I can find no mention of this in the main text. This needs to be clarified in the main text, not just the methods, otherwise it is misleading! And on my previous point of looking at whether associations are driven by LoF or missense variants - yes, it is the case that rare missense variants can cause LoF, however, if there are no identified LoF variants and instead the association is driven by missense variants, this could suggest an alternative mechanism.

We have clarified the main text in the results to read:

To further validate this hypothesis, we performed a gene burden test by aggregating rare predicted loss-of-function (pLOF) protein-coding variants in the PMBB and UKB for each significant uORF-PheWAS association. Rare (MAF \leq 0.1%) pLOF variants were defined as frameshift insertions/deletions, gain/loss of stop codon, or disruption of canonical splice site dinucleotides. Predicted deleterious rare (MAF \leq 0.1%) missense variants were defined as those with Rare Exonic Variant Ensemble Learner (REVEL) scores \geq 0.5 and included in the set of pLOF protein-coding variants for gene-burden analyses.

- Relating to my previous point "How does the enrichment in Figure 1c compare to stop codons used in canonical coding sequences?" I do not think it is out of scope of the manuscript to at least note how the proportions of different stop codons in uORFs compares to what is seen for CDS stop codons - i.e. are they on average weaker?

In canonical protein-coding regions of the genome the frequency of stop codon usage has been observed to decline in following the pattern UGA, UAA, and UAG⁴. We have added text to our results section acknowledging this difference between stop codon usage in uORFs compared to canonical protein-coding sequences.

References

1. Yang, Y.-L. *et al.* Expression and prognostic significance of the apoptotic genes BCL2L13, Livin, and CASP8AP2 in childhood acute lymphoblastic leukemia. *Leuk. Res.* **34**, 18–23 (2010).
2. Jensen, S. A. *et al.* Bcl2L13 is a ceramide synthase inhibitor in glioblastoma. *Proc. Natl. Acad. Sci. U. S. A.* **111**, 5682–5687 (2014).
3. García-Cazorla, À. *et al.* Impairment of the mitochondrial one-carbon metabolism enzyme SHMT2 causes a novel brain and heart developmental syndrome. *Acta Neuropathol.* **140**,

971–975 (2020).

4. Sun, J., Chen, M., Xu, J. & Luo, J. Relationships among stop codon usage bias, its context, isochores, and gene expression level in various eukaryotes. *J. Mol. Evol.* **61**, 437–444 (2005).